# Transition-constant Normalization for Image Enhancement

**Jie Huang**[1]*, **Man Zhou**[1,2]*, **Jinghao Zhang**[1] , **Gang Yang**[1], **Mingde Yao**[1],
**Chongyi Li**[3], **Zhiwei Xiong**[1], **Feng Zhao**[1]†
[1]University of Science and Technology of China, China
[2]Nanyang Technological University, Singapore
[3]Nankai University, China
{hj0117, manman, jhaozhang, mdyao, yg1997}@mail.ustc.edu.cn,
lichongyi25@gmail.com, {zwxiong, fzhao956}@ustc.edu.cn,

## Abstract

Normalization techniques that capture image style by statistical representation have become a popular component in deep neural networks. Although image enhancement can be considered as a form of style transformation, there has been little exploration of how normalization affect the enhancement performance. To fully leverage the potential of normalization, we present a novel Transition-Constant Normalization (TCN) for various image enhancement tasks. Specifically, it consists of two streams of normalization operations arranged under an invertible constraint, along with a feature sub-sampling operation that satisfies the normalization constraint. TCN enjoys several merits, including being parameter-free, plug-and-play, and incurring no additional computational costs. We provide various formats to utilize TCN for image enhancement, including seamless integration with enhancement networks, incorporation into encoder-decoder architectures for downsampling, and implementation of efficient architectures. Through extensive experiments on multiple image enhancement tasks, like low-light enhancement, exposure correction, SDR2HDR translation, and image dehazing, our TCN consistently demonstrates performance improvements. Besides, it showcases extensive ability in other tasks including pan-sharpening and medical segmentation. The code is available at *https://github.com/huangkevinj/TCNorm*.

## 1 Introduction

Image enhancement is an important task in machine vision, which aims to improve the quality of low-visibility images captured under unfavorable light conditions (i.e., low light) by adjusting contrast and lightness. The last decades have witnessed quantities of approaches designed for image enhancement based on various hand-crafted priors [1, 2, 3, 4, 5]. However, the complex and variant adjustment procedures make it a challenging group of tasks. In addition to the common low-light image enhancement, efforts have also been directed toward solving image enhancement-like tasks, including exposure correction, image dehazing, and SDR2HDR translation.

Very recently, the deep-learning paradigm exhibits remarkable success in the image enhancement field than traditional methods [6, 7, 8]. Despite the progress, most of them focus on roughly constructing complex deep neural architectures and have not fully explored the intrinsic characterizes of lightness in networks. In fact, the lightness variants could bring difficulties to their learning procedures. This

---

*Both authors contributed equally to this research.
†Corresponding author.

37th Conference on Neural Information Processing Systems (NeurIPS 2023).

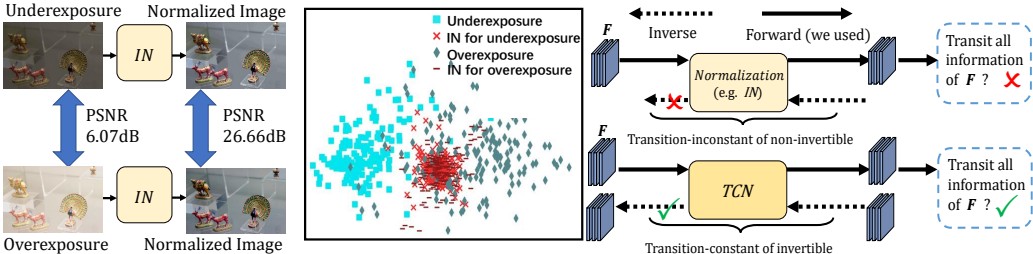

(a) Apply normalization for different exposure images.     (b) Transition-constant property of normalization.

Figure 1: In (a), the instance normalization (IN) captures lightness consistency representations across exposures, thus bridging their distribution gaps as shown in t-SNE. In (b), normalization techniques meet transition-inconstant problem, while proposed TCN exhibits differently with invertible ability.

motivates us to delve into the working mechanism of current neural networks that learn lightness adjustment and prescribe the right medicine customized for image enhancement.

On the other hand, normalization family such as batch and instance normalization, is specially designed for promoting the learning procedure for deep networks. It involves computing statistical representation and normalizing the corresponding distribution, which has been shown to capture image style through statistical representation [9]. Image enhancement, which aims to restore lightness-corrupted images to their normal versions, can be viewed as a style transformation inherently linked to normalization techniques (see Fig. 1 (a)). However, existing methods have rarely explored the potential of the normalization technique. Inspired by the inborn connection, we thus focus on developing the normalization technique tailored for image enhancement.

In this work, we propose a novel operation called transition-constant normalization (TCN) for image enhancement tasks. The TCN operation aims to normalize partial representations to ensure consistent learning while preserving constant information for image reconstruction. As illustrated in Fig. 1 (b), we construct the TCN within an invertible format to enable seamless information transmission to subsequent layers for reconstructing enhanced results. The TCN is designed with two key rules (see Fig. 2): (1) We organize the operations in the normalization layer into two streams, following the principle of invertible information transmission, thereby maintaining constant information transition. (2) We incorporate a subsampling operation that divides the features into two streams with consistent statistical properties. One stream provides the statistics for normalizing the other stream, satisfying the normalization requirement. Notably, the TCN requires no parameters, making it a convenient and orthogonal addition to existing enhancement architectures for improving their performance.

To facilitate its application, we present multiple usage formats for the TCN in image enhancement: (1) Integration into existing enhancement networks, allowing for seamless incorporation and performance improvement; (2) Plug-in capability in encoder-decoder architectures for downsampling and recomposition of information; (3) Construction of a lightweight architecture based on TCN, striking a balance between performance and computational cost. Through extensive experiments across various image enhancement tasks, we consistently observe performance gains by integrating our TCN.

The contributions of this work are summarized as follows: 1) We present a novel perspective on image enhancement using a dedicated normalization technique. This technique enhances the learning of lightness adjustments by modeling consistent normalized features, while ensuring their complementarity for reconstructing the results. 2) We construct the Transition-constant Normalization (TCN) by organizing normalization operations to satisfy the invertible mechanism, ensuring constant feature normalization and information transition. 3) Our proposed TCN is compatible with existing enhancement architectures, allowing for convenient integration and performance improvement. Furthermore, we can derive multiple implementation formats for TCN and explore its applicability in various tasks, highlighting its potential for wide-ranging applications.

## 2 Related Work

**Image enhancement tasks.** Image enhancement tasks aim to improve the quality of low-visibility images by adjusting lightness and contrast components (e.g., illumination, color, and dynamic range).

Recent years have witnessed rapid development in the related areas [10, 11, 12, 13]. For low-light image enhancement, algorithms are designed to enhance the visibility of images captured under low-light conditions [14, 15, 16, 17, 18, 19, 7, 20, 21, 22, 23]. In the exposure correction task, methods are focused on correcting both underexposure and overexposure to normal exposure [15, 24, 25, 26, 27]. For SDR2HDR translation, this task aims to design methods to convert images from a low-dynamic range to a high-dynamic range [28, 29, 30, 31, 32]. While in image dehazing, this task requires methods to enhance the contrast and recover the color shift problems [33, 34, 35, 36, 37]. To this end, image enhancement tasks cover variant scenes and remain challenges to be solved.

**Normalization techniques.** Normalization techniques have been studied for a long time [38, 39, 40, 41]. Batch Normalization (BN) [38] normalizes the features along the batch dimension that stabilizes the optimization procedure. Instance Normalization (IN) [9] focuses on normalizing the instance-level statistics of features, which has been widely employed in style transfer tasks [42, 43]. Some other variants of normalization, including Layer Normalization (LN) [39], Group Normalization (GN) [44], and Position Normalization (PN) [45] have been proposed for facilitating the application of networks.

## 3 Method

In this section, we first briefly revisit the normalization techniques and then detail the design and mechanism of the proposed TCN. Finally, we present the variants of the TCN as implementation.

### 3.1 Preliminaries

Given a batch of features $x \in \mathbb{R}^{N \times C \times H \times W}$, where N, C, H and W represent batch size, channel numbers, the spatial height and width, respectively. Let $x_{ncij}$ and $\hat{x}_{ncij}$ denote a pixel before and after normalization, where $n \in [1, N], c \in [1, C], i \in [1, H], i \in [1, W]$. Without taking considering into affined parameters, we can express normalization operation as:

$$\hat{x}_{ncij} = \text{Norm}(x_{ncij}) = \frac{x_{ncij} - \mu_k}{\sqrt{\sigma_k^2 + \epsilon}}, \tag{1}$$

where $\mu_k$ and $\sigma_k$ denote the feature mean and standard deviation, $\epsilon$ is a small constant to preserve numerical stability. $k \in \{\text{IN, BN, LN, GN}\}$ is to distinguish different normalization formats. Within the above normalization family, the calculation of $\mu_k$ and $\sigma_k$ is different and are expressed as:

$$\mu_k = \frac{1}{|I_k|} \sum_{n,c,i,j \in I_k} x_{ncij}, \quad \sigma_k = \sqrt{\frac{1}{|I_k|} \sum_{n,c,i,j \in I_k} (x_{ncij} - \mu_k)^2}, \tag{2}$$

1) **IN**: $I_k : I_{IN} = \{(i, j)|i \in [1, H], j \in [1, W]\}$;
2) **BN**: $I_k : I_{BN} = \{(n, i, j)|n \in [1, N], i \in [1, H], j \in [1, W]\}$;
3) **LN**: $I_k : I_{LN} = \{(c, i, j)|c \in [1, C], i \in [1, H], j \in [1, W]\}$;
4) **GN**: $I_k : I_{GN} = \{(c, i, j)|c \in [g, g + C/G], i \in [1, H], j \in [1, W]|g \in [1, G]\}$.

where $I_k$ is a set of pixels, $|I_k|$ denotes the number of pixels and $G$ is the group division.

Within deep neural networks, Eq. 1 is often affined with scaling and shifting parameters $\alpha$ and $\beta$:

$$\hat{x}_{ncij} = \alpha \cdot \text{Norm}(x_{ncij}) + \beta = \alpha \frac{x_{ncij} - \mu_k}{\sqrt{\sigma_k^2 + \epsilon}} + \beta. \tag{3}$$

It is well-known that lightness can be considered as a kind of style and Instance normalization can facilitate consistent style information [9] and image enhancement network optimization due to bridging the gap of different lightness representations.

**Verifying normalization effect for lightness**. Given an image $x_a$ and its lightness-adjusted version $x_b$, the relationship between $x_a$ and $x_b$ can be expressed in correction procedure [46, 47, 48] as:

$$x_b = \Lambda x_a^\gamma, \tag{4}$$

where $\Lambda$ is a linear transformation and $\gamma$ is for global non-linear adaption and is close to 1 when content is not severely changed. Therefore, the p-norm distance between their normalized versions is:

$$||\text{f}(x_a) - \text{f}(x_b)||_p = |\frac{x_a - \mu_a}{\sigma_a} - \frac{\Lambda x_a^\gamma - \mu_b}{\sigma_b}||_p \approx |\frac{x_a - \mu_a}{\sigma_a} - \frac{\Lambda x_a - \Lambda \mu_a}{\Lambda \sigma_a}||_p \ll ||x_a - \Lambda x_a^\gamma||_p. \tag{5}$$

Therefore, normalization reduces the distance of different lightness, which is also validated in Fig. 1(a). However, normalizing the statistics itself often blocks the information flow, which hinders the network from reconstructing the final results [49] and we describe it as follows.

**Transition-inconstant of Normalization.** Referring to $\mathrm{Norm}(\cdot)$ in the Eq. 1 as $f$, its Jacobian matrix is expressed as:

$$J_f(x) = \begin{bmatrix} \frac{\partial \hat{x}_0}{x_0} & \cdots & \frac{\partial \hat{x}_0}{x_0} \\ & \ddots & \vdots \\ \frac{\partial \hat{x}_0}{x_0} & & \frac{\partial \hat{x}_0}{x_0} \end{bmatrix}_{m \times m} = \begin{bmatrix} \frac{1}{\sigma_k} & \cdots & \frac{1}{\sigma_k} \\ & \ddots & \vdots \\ \frac{1}{\sigma_k} & & \frac{1}{\sigma_k} \end{bmatrix}_{m \times m} \tag{6}$$

Therefore, the calculation of the above Jacobian matrix is $\mathbf{det}\,(J_f(x)) = 0$, denoting the normalization operation is not invertible and resulting in transition-inconstant. Meanwhile, in practice, previous works have demonstrated that IN would lead to severe information loss [42] and huge representation ability changes [50, 51], while LN and BN can keep almost all of the original information representation ability. However, IN is more suitable than BN and LN for image enhancement tasks due to its strong capability of capturing and affecting style information, which is crucial for image enhancement. To this end, the main goal of this paper is to introduce a new mechanism that enables the IN can keep the information representation ability for image enhancement.

### 3.2 Transition-constant Normalization (TCN)

Based on the above analysis, we aim to refresh the normalization technique to enable it to transmit information constantly while normalizing the features. To this end, we introduce the TCN as shown in Fig. 2, which is free of parameters and is convenient to implement. Since IN can normalize different lightness effectively and thus is useful for image enhancement, we design the TCN based on the IN as its default implementation format in this paper.

**Operation description.** We construct the TCN by applying the normalization operations with a two streams flow design with subsampled features, where one stream provides the statistical information for normalizing another stream. Specifically, the feature $F \in \mathbb{R}^{B \times C \times H \times W}$ is firstly subsampled to $F_s \in \mathbb{R}^{B \times 4C \times \frac{H}{2} \times \frac{W}{2}}$ according to the unshuffle operation [52] as shown in Fig. 2 (a), which is:

$$F_s^{ab} = F[:,:,a::,b::], \quad a,b \in \{0,1\}, \tag{7}$$

where $a$ and $b$ denote the subsampling index. We divide $F_s$ into two features $F_1$ and $F_2$ with two groups according to the sampling index $(i,j)$ as:

$$F_1 = \mathrm{Concat}(F_s^{01}, F_s^{10}), F_2 = \mathrm{Concat}(F_s^{00}, F_s^{11}) \tag{8}$$

where $\mathrm{Concat}(\cdot,\cdot)$ denotes the concatenate operation along the channel dimension.

Then, we calculate the mean $\mu_2$ and standard deviation $\sigma_2$ of one stream feature $F_2$ in IN format, which are derived by setting $I_k$ in Eq. 2 as $I_{IN}$:

$$\mu_2 = \frac{1}{\mathrm{HW}} \sum_{i \in [1,\mathrm{H}], j \in [1,\mathrm{W}]} F_{ij}, \quad \sigma_2 = \sqrt{\frac{1}{\mathrm{HW}} \sum_{i \in [1,\mathrm{H}], j \in [1,\mathrm{W}]} (F_{ij} - \mu_2)^2}. \tag{9}$$

These statistics are utilized to normalize the feature $F_1$ as the output in this stream:

$$\hat{F}_1 = \frac{F_1 - \mu_2}{\sqrt{\sigma_2^2 + \epsilon}}. \tag{10}$$

Next, we subtract the feature $F_2$ and $\hat{F}_1$ and obtain the output of another stream:

$$\hat{F}_2 = F_2 - \hat{F}_1 = F_2 - \frac{F_1 - \mu_2}{\sqrt{\sigma_2^2 + \epsilon}}. \tag{11}$$

Finally, the two stream features are sampled with pixel shuffle operation to the original resolution with the shape of $[\mathrm{N}, \frac{\mathrm{C}}{2}, \mathrm{H}, \mathrm{W}]$. They are further concatenated in the channel dimension as $\hat{F} \in \mathbb{R}^{B \times C \times H \times W}$, which is the output of the TCN. This procedure is expressed as:

$$\hat{F} = \mathrm{Pixshuffle}(\mathrm{Concat}(\hat{F}_1, \hat{F}_2)), \tag{12}$$

where $\text{Pixshuffle}(\cdot)$ denotes the pixel shuffle operation [53]. We verify the above procedures satisfy the transition-constant and normalization ability as below, respectively.

**1) Verify the transition-constant ability.** We validate the above two-stream design satisfies the invertible procedure and thus is transition-constant. To this end, Eq. 10 and Eq. 11 are re-written as:

$$\begin{aligned}
\hat{F}_1 &= (F_1 - M(F_2)) \oslash S(F_2), \\
\hat{F}_2 &= F_2 - \hat{F}_1,
\end{aligned} \tag{13}$$

where $S(\cdot)$ and $M(\cdot)$ denote standard deviation and mean functions, $\oslash$ denotes element division.

Inspired by the proof in RealNVP's [54] transformation, we need to calculate the Jacobian matrix of Eq. 13 (denote it as $g$), which is more intuitively written as:

$$\begin{aligned}
\hat{F}_1 &= (F_1 - M(F_2)) \oslash S(F_2), \\
\hat{F}_2 &= F_2 - F_1 \oslash S(F_2) + M(F_2) \oslash S(F_2),
\end{aligned} \tag{14}$$

We derive its Jacobian matrix (detailed in the supplementary) as:

$$J_g = \begin{bmatrix} \frac{\partial \hat{F}_1}{\partial F_1} & \frac{\partial \hat{F}_1}{\partial F_2} \\[2mm] \frac{\partial \hat{F}_2}{\partial F_1} & \frac{\partial \hat{F}_2}{\partial F_2} \end{bmatrix} = \begin{bmatrix} \frac{1}{\partial S(F_2)} & 0 \\[2mm] -1 & 1 \end{bmatrix} \tag{15}$$

Here, the above Jacobian matrix is further calculated as:

$$\mathbf{det}(J_g) = \frac{1}{\partial S(F_2)} \neq 0 \tag{16}$$

Upon the $\mathbf{det}(J_g) \neq 0$, it indicates that $J_g$ is full rank, verifying the invertible property of TCN and further the transition-constant ability. To highlight, the TCN is an invertible function and would not block information flow, leading to the information transition constant for image reconstruction. We further present the relation of the TCN and the invertible operation more directly in the supplementary.

| Formats | Redefine $I_k$ to Eq. 9 as |
|---|---|
| TCN (IN) | $I_{IN}$ (default $I_k$ of Eq. 9) |
| TCN (BN) | $I_{BN} = \{(n,i,j)\|n \in [1,N], i \in [1,H], j \in [1,W]\}$ |
| TCN (LN) | $I_{LN} = \{(c,i,j)\|c \in [1,C], i \in [1,H], j \in [1,W]\}$ |
| TCN (GN) | $I_{GN} = \{(c,i,j)\| c \in [g, g+C/G], i \in [1,H], j \in [1,W]\|g \in [1,G]\}$ |

Table 1: The TCN family with different $\mu_2, \sigma_2$ calculation in Eq. 9.

**2) Verify the normalization ability.** The normalization ability of the TCN is guaranteed by the pixel unshuffle operation in Eq. 7, leading to the same statistics of $F_s^{ab}$ [55, 56]. Therefore, we have $\mu_2 \approx \mu_1, \sigma_2 \approx \sigma_1$, and the Eq. 10 is thus converted to:

$$\hat{F}_1 = \frac{F_1 - \mu_2}{\sqrt{\sigma_2^2 + \epsilon}} \approx \frac{F_1 - \mu_1}{\sqrt{\sigma_1^2 + \epsilon}}. \tag{17}$$

Therefore, it has the same format as Eq. 1, demonstrating that the operation in Eq. 10 has the normalization ability as the IN. Further, we verify the above rules by the toy experiment (see Sec. 4.1) in Fig. 4 and Fig. 5.

**Discussion.** The core of the TCN is the statistic calculation manner of $\mu_2$ and $\sigma_2$ in Eq. 9 which can be generalized in a unified calculation manner in Eq. 2 and derive from other normalization formats of TCN, shown in Table 1. Note that, although GN and LN would less affect information representation ability [50], we experimentally find introducing the transition-constant design would improve their performance in the supplementary. We provide more discussions in the supplementary.

### 3.3 Variants of TCN for Image Enhancement

Upon the above principles of TCN, we provide the following implementation variants within image enhancement task.

**The original TCN.** We construct the original TCN (see Fig. 2 (a)) for image enhancement based on calculating statistics in Eq. 9 of IN format, which is plug-and-play for networks.

**The affined TCN.** We extend the original TCN by introducing affined parameters $\alpha$ and $\beta$ to the normalization procedure, resulting in the affined TCN (Fig. 2 (b)). We incorporate learnable shifting

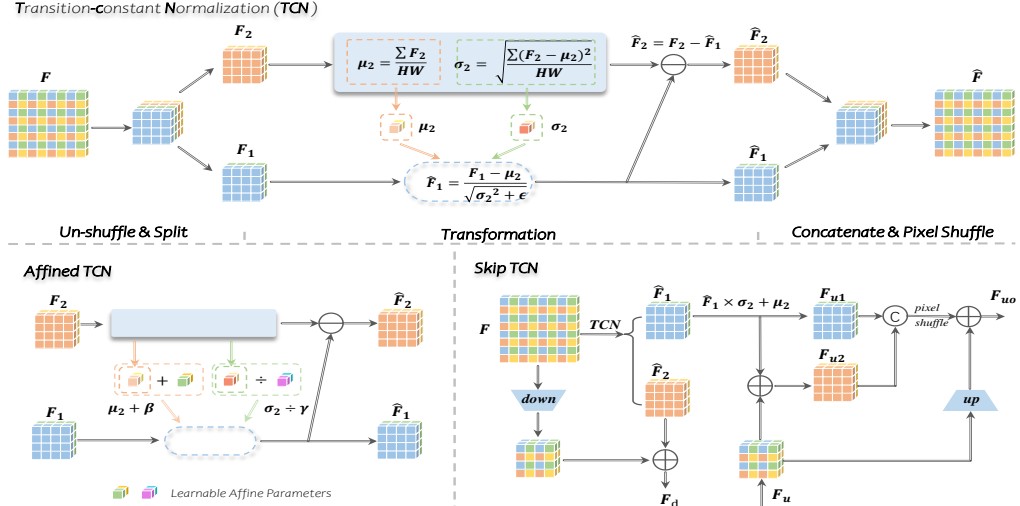

Figure 2: The illustration of the TCN operation and other TCN variants for image enhancement.

parameter $\beta$ and scaling parameter $\gamma$ into $\mu_2$ and $\sigma_2$ in Eq. 9:

$$\mu'_2 = \mu_2 + \beta, \quad \sigma'_2 = \frac{\sigma_2}{\gamma} \tag{18}$$

where $\mu'_2$ and $\sigma'_2$ represent the affined statistics. Then, we substitute Eq. 18 to Eq. 1:

$$\hat{F}'_1 = \frac{F_1 - \mu'_2}{\sqrt{\sigma'^2_2 + \epsilon}} = \frac{F_1 - \mu_2 - \beta}{\sqrt{\frac{\sigma_2^2}{\gamma^2} + \epsilon}} \tag{19}$$

Since $\epsilon$ is a small constant near to 0, the Eq. 19 can be approximated as:

$$\hat{F}'_1 \approx \gamma \frac{F_1 - \mu'_2 - \beta}{\sqrt{\sigma'^2_2 + \epsilon}} = \gamma \frac{F_1 - \mu'_2}{\sqrt{\sigma'^2_2 + \epsilon}} + \beta', \quad \beta' = \frac{-\gamma\beta}{\sqrt{\sigma'^2_2 + \epsilon}}. \tag{20}$$

Eq.20 shares a format similar to the affined normalization, with $\gamma$ and $\beta'$ as the learnable scaling and shifting parameters in Eq. 3. The affined TCN seamlessly integrates into image enhancement networks, serving as a plug-and-play solution. Notably, it maintains the information transition-constant property, as discussed in detail in the supplementary material.

**The skip TCN.** From Eq.10 and 11, the TCN generates two types of features: a domain-invariant lightness consistent feature $\hat{F}_1$ and a domain-variant lightness inconsistent feature $\hat{F}_2$. Previous studies [57, 58] have demonstrated the effectiveness of incorporating the domain-variant component into deep encoder-decoder networks while skipping the domain-invariant component to the decoder layer. In this work, we propose the skip TCN architecture, illustrated in Fig. 2 (c).

Given a feature $F$ in an encoder layer, we convey its lightness inconsistent feature $\hat{F}_2$, obtained from Eq. 10, to the downsampled deeper layer that derives $F_d$ using the following expression:

$$F_d = \text{Down}_2(\text{F}) + \hat{\text{F}}_2, \tag{21}$$

where $\text{Down}_2$ means downsampling with a factor of 2. While for the lightness consistent feature $\hat{F}_1$ derived in Eq. 11, we skip it to the corresponding decoder layer feature $F_u$ with the statistic $\mu_2$ and $\sigma_2$ derived in Eq. 9. We integrate them by inverting the operation of Eq. 10 and Eq. 11:

$$\begin{aligned} F_{u1} &= \hat{F}_1 \cdot \sigma_2 + \mu_2, \quad F_{u2} = \hat{F}_1 + F_u, \\ F_{uo} &= \text{Up}_2(\text{F}_\text{u}) + \text{Pixshuff}(\text{Concat}(\text{F}_{\text{u1}}, \text{F}_{\text{u2}})), \end{aligned} \tag{22}$$

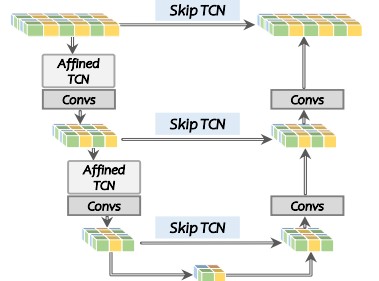

Figure 3: The overview of the TCN-Net.

Here, the upsampling operation with a factor of 2 is denoted as $\text{Up}_2$, and $F_{uo}$ represents the skip TCN result. The skip TCN prioritizes the processing of the lightness component while preserving lightness invariant features, mitigating learning difficulties. Further discussion is provided in the supplementary material.

**Construct a very efficient TCN-based Network.** We introduce TCN-Net, an efficient network architecture depicted in Fig. 3, which combines affined TCN and skip TCN. This framework adopts an encoder-decoder-based architecture with vanilla convolution blocks to depict the effectiveness of the TCN. Further details and discussions are available in the supplementary material.

# 4 Experiments

In this section, we validate the effectiveness and scalability of our proposed TCN on various image enhancement tasks. We provide more experimental results in the supplementary material.

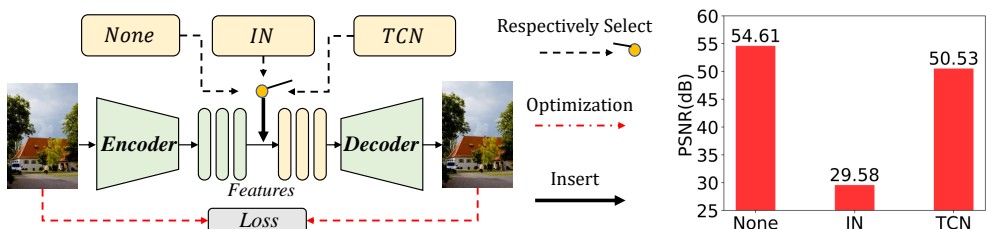

Figure 4: Toy experiment of self-reconstruction. The left is the setting of toy experiments with inserting different operations, and the right presents the self-reconstruction PSNR of testing images.

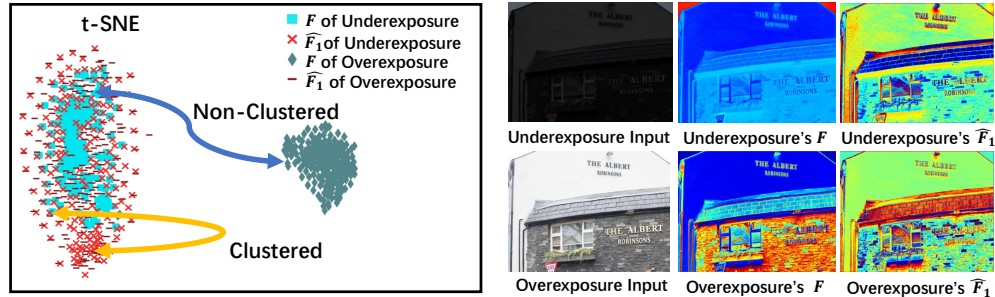

Figure 5: Feature visualization of toy experiment. In left and right parts, we show the feature in the TCN of underexposure and overexposure samples when testing them with inserting TCN in Fig. 4.

## 4.1 Toy Experiment

To illustrate the proposed TCN has the ability to normalize the features while transition-constant, we introduce a toy experiment as shown in Fig. 4 : we construct an encoder-decoder-based architecture for reconstructing the input image, where the TCN and other normalization formats are inserted between the encoder and decoder as different versions. Then we train this self-reconstruction architecture on 1000 samples from MIT-FiveK dataset [59] until its convergence and test the self-reconstruction effect on another 100 samples from the same dataset. The quantitative results in the right of Fig. 4 indicate that our TCN reconstructs the input image better than directly inserting IN, demonstrating the information transition-constant property. Furthermore, we test 100 underexposure samples and 100 overexposure samples from the SICE dataset, and we provide the feature distribution of $F$ and $\hat{F}_1$ (input and normalized output of the TCN) in the left part of Fig. 5 , as well as feature maps in the right part of Fig. 5. The different exposure features processed by the TCN get to be

| Settings | #Param | Flops (G) | LOL | Huawei | FiveK |
|---|---|---|---|---|---|
| DRBN (Baseline) [15] | 0.53M | 39.71 | 19.95/0.7712 | 20.64/0.6136 | 22.11/0.8684 |
| +IN | 0.53M | 39.71 | 20.73/0.7986 | 21.01/0.6200 | 22.93/0.8727 |
| +Original TCN | 0.53M (+0) | 39.71 (+0) | 21.15/**0.8190** | 21.12/**0.6242** | **23.98**/0.8851 |
| +Affined TCN | 0.53M (+0) | 39.71 (+0) | 21.29/0.8167 | 21.04/0.6231 | 23.92/**0.8858** |
| +Skip TCN | 0.53M (+0) | 39.77 (+0.07) | **21.52**/0.8271 | **21.15**/0.6195 | 23.82/0.8832 |
| SID (Baseline) [14] | 7.40M | 51.06 | 20.85/0.7845 | 19.68/0.6050 | 21.49/0.8425 |
| +IN | 7.40M | 51.06 | 20.51/0.7858 | 20.09/0.6034 | 21.75/0.8453 |
| +Original TCN | 7.40M (+0) | 51.06 (+0) | 21.43/0.7913 | 20.53/0.6067 | 23.11/0.8581 |
| +Affined TCN | 7.40M (+0) | 51.06 (+0) | 21.35/0.7867 | 20.62/0.6077 | 23.20/0.8624 |
| +Skip TCN | 7.41M (+0.01) | 51.42 (+0.36) | **21.92/0.8056** | **20.76/0.6083** | **23.61/0.8704** |
| TCN-Net | 0.012M | 0.97 | 22.08/0.7895 | 20.99/0.6121 | 23.47/0.8663 |

Table 2: Comparison over low-light image enhancement in terms of PSNR/MS-SSIM.

| Settings | MSEC | SICE |
|---|---|---|
| DRBN (Baseline) [15] | 19.52/0.8309 | 17.65/0.6798 |
| +IN | 21.98/0.8463 | 20.15/0.6947 |
| +Original TCN | 22.37/0.8533 | 20.74/0.7133 |
| +Affined TCN | 22.41/0.8504 | **20.85/0.7192** |
| +Skip TCN | **22.48/0.8572** | 20.65/0.7159 |
| SID (Baseline) [14] | 19.04/0.8074 | 18.15/0.6540 |
| +IN | 21.36/0.8373 | 19.81/0.6667 |
| +Original TCN | 22.31/0.8522 | 20.51/0.6745 |
| +Affined TCN | **22.43**/0.8542 | **20.68**/0.6757 |
| +Skip TCN | 22.36/**0.8603** | 20.64/**0.6852** |
| TCN-Net | 22.19/0.8480 | 20.72/0.7024 |

Table 3: Comparison over exposure correction.  Figure 6: Training PSNR on exposure correction.

clustered, demonstrating the normalization ability of the TCN for extracting the lightness-consistence representation of different samples. We provide more discussions in the supplementary material.

## 4.2 Experimental Settings

**Low-light Image Enhancement.** Following previous works [60, 61], we employ three widely used datasets for evaluation, including LOL dataset [7], Huawei dataset [60] and MIT-FiveK dataset [59]. We employ two different image enhancement networks, DRBN [15] and SID [14] as baselines.

**Exposure Correction.** Following [62], we adopt MSEC dataset [24] and SICE dataset [63] for evaluations. The above two architectures, i.e., DRBN [15] and SID [14] are regarded as baselines.

**SDR2HDR Translation.** Following [30], we choose the SRITM dataset [31] and HDRTV dataset [30] for evaluation. We employ the structures of NAFNet [64] with its three basic units as the baseline in the experiments.

**Image Dehazing.** Following [33], we employ the RESIDE dataset [65] consisting of Indoor and Outdoor parts for evaluations. We adopt the network of PFFNet [66] as the baseline for validation.

## 4.3 Implementation Details

Since there exist three TCN formats in Sec. 3.3, we respectively integrate them into the baseline to conduct experiments. For comparison, we perform the experiments of baseline networks and the integration of the IN operation. Additionally, the TCN-Net in Sec. 3.3 is also performed in experiments. We train all baselines and their integrated formats following the original settings, and our TCN-Net until it converges. More implementation details are provided in the supplementary.

| Settings | SRITM | HDRTV | | Settings | Indoor | Outdoor |
|---|---|---|---|---|---|---|
| NAFNet (Baseline) [64] | 33.44/**0.9537** | 36.49/0.9706 | | PFFNet (Baseline) [66] | 21.74/0.8452 | 24.47/0.9274 |
| +IN | 33.62/0.9491 | 36.62/0.9683 | | +IN | 23.13/0.8583 | 25.61/0.9309 |
| +Original TCN | **33.69**/0.9505 | **36.94**/0.9712 | | +TCN | 23.57/0.8635 | 25.63/0.9311 |
| +Affined TCN | 33.65/0.9495 | 36.55/0.9716 | | +Affined TCN | **23.71**/0.8652 | **25.84**/0.9312 |
| +Skip TCN | 33.51/0.9513 | 36.64/**0.9720** | | +Skip TCN | 23.21/**0.8708** | 25.63/**0.9315** |
| TCN-Net | 32.48/0.9439 | 36.78/0.9744 | | TCN-Net | 24.06/0.8645 | 23.72/0.8572 |

Table 4: Comparison over SDR2HDR translation.  Table 5: Comparison over image dehazing.

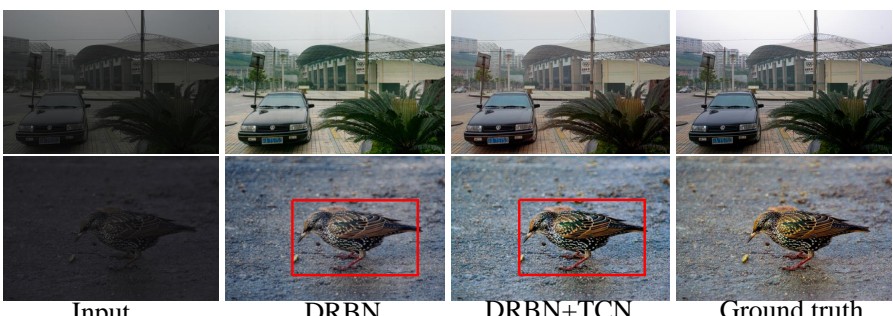

|  Input  |  DRBN  |  DRBN+TCN  |  Ground truth  |

Figure 7: The visual comparison of low-light image enhancement on the MIT-FiveK dataset.

## 4.4 Comparison and Analysis

**Quantitative Comparison.** The model comparisons are conducted over different configurations, as illustrated in the implementation details. We present the quantitative results from Table 2 to Table 5, where the best and second-best results are highlighted in bold and underlined. As can be seen, almost all formats of the TCN that we incorporate have improved the performance across the datasets in all tasks, validating the effectiveness of our method. Specifically, integrating variants of TCN helps improve the training performance of baseline as shown in Fig. 6. In contrast, naively integrating the IN could not always bring performance improvement (i.e., the results of SID in Table 2). All the above results suggest the effectiveness of our proposed method without introducing any parameters. Moreover, the proposed TCN-Net achieves effective performance with efficiency. All the above evaluations prove the convenience of applying the TCN in image enhancement tasks.

**Qualitative Comparison.** We report the visual results of low-light image enhancement on the MIT-FiveK dataset [59] due to the limited space. As shown in Fig. 7, the integration of the TCN leads to a more visually pleasing effect with less lightness and color shift problems compared with the original baseline. We provide more visual results in the supplementary material.

## 4.5 Extensive Applications

The TCN can also be applied to other machine vision tasks that demonstrate its extensibility. Since TCN is proposed to extract lightness (a kind of style) invariant feature while keeping information transition-constant, we introduce another two tasks that are also related to style information, including pan-sharpening and medical segmentation. For pan-sharpening, it aims to fuse two style images, and we hope TCN can extract their invariant information with information preserving; For medical segmentation, there often exists a style domain gap between training and testing sets.

**Extension on medical segmentation.** We apply the TCN on the UNet [67] and AttUNet [68] in the medical segmentation task. We train the baseline and its integrated version on the heart segmentation task of Medical Segmentation Decathlon challenge dataset [69]. As shown in Table 6, our TCN improves and keeps the performance of U-Net and Att-Unet, respectively, while IN brings a significant performance drop. The results suggest the scalability of the TCN compared with the IN.

**Extension on pan-sharpening.** We apply the original TCN to the GPPNN [70] and PANNet [71] baselines in the pan-sharpening task, which is a common task in guided image super-resolution. We integrate it when extracting pan and multi-spectral features, and experimental results on WorldView II dataset [72, 73] in Fig. 8 suggest the effectiveness of the TCN.

| Settings | Dice ↑ | HD95↓ |
|---|---|---|
| UNet(baseline) | 0.9162 | **3.9188** |
| UNet(+IN) | 0.9171 | 7.7305 |
| UNet(+TCN) | **0.9204** | 4.0171 |
| AttUNet(baseline) | 0.9182 | **3.5453** |
| AttUNet(+IN) | **0.9193** | 6.8549 |
| AttUNet(+TCN) | 0.9180 | 3.6241 |

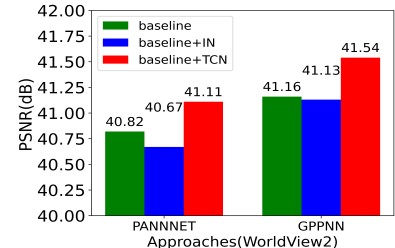

Table 6: Comparison over medical segmentation in terms of dice and HD95.

Figure 8: Comparison over pan-sharpening for GPPNN and PANNet.

## 5  Limitation and Discussion

Firstly, we validate the effectiveness of TCN in image enhancement tasks, while the investigation of applying TCN to other image restoration tasks will be explored in the future, such as the all-in-one image restoration task that meets similar challenges like image enhancement tasks, which has been pointed in some related works [74, 75]. Second, dedicated to image enhancement tasks, we mainly discuss the IN format of the TCN. However, other normalization formats can be future explored for other tasks. Moreover, the design formats of the TCN could inspire some areas that also require transition-constant, such as image fusion tasks [76]. Finally, the TCN could introduce very few computation burdens although it is free of parameters, which is negligible compared with its bring performance improvement. Note that the focus of this work is beyond introducing a plug-and-play operation to existing networks for performance gain. The introduced TCN can be a new choice of normalization and feature disentanglement, which excavate consistent representations while preserving information when developing a new model that requires this property.

## 6  Conclusion

In this paper, we introduce a new perspective that develops the normalization technique tailored for image enhancement approaches. We propose the TCN that transits the information constantly with the invertible constraint, meanwhile, it keeps the normalization ability for capturing lightness consistence representations. The proposed TCN is a general operation that can be integrated into existing networks without introducing parameters. Extensive experiments demonstrate the effectiveness and scalability of applying the TCN and its variants in various image enhancement tasks.

## Broader Impact

Image enhancement is an important task that improves the quality of these images, exhibiting a high value of research and application. Our method introduces a normalization operation with information transition-constant property, which shows promising results that improve the learning ability of networks for image enhancement tasks conveniently. However, there could be negative effects brought by the proposed methodology. For example, some people may prefer the image with a dim light effect, which would be eliminated by the image enhancement algorithm. In these cases, it is suggested to combine the users' preferences to achieve customized image enhancement effects.

## Acknowledgements.

This work was supported by the JKW Research Funds under Grant 20-163-14-LZ-001-004-01, and the Anhui Provincial Natural Science Foundation under Grant 2108085UD12. We acknowledge the support of GPU cluster built by MCC Lab of Information Science and Technology Institution, USTC.

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
