# "Transition-constant Normalization for Image Enhancement" Supplementary Material

**Jie Huang**[1][*] **Man Zhou**[1,2][*] **Jinghao Zhang**[1] , **Gang Yang**[1]**, Mingde Yao**[1],
**Chongyi Li**[3]**, Zhiwei Xiong**[1]**, Feng Zhao**[1][†]
[1]University of Science and Technology of China, China
[2]Nanyang Technological University, Singapore
[3]Nankai University, China
{hj0117, manman, jhaozhang, mdyao, yg1997}@mail.ustc.edu.cn,
lichongyi25@gmail.com, {zwxiong, fzhao956}@ustc.edu.cn,

This supplementary document is organized as follows:

Sec. 1 provides more results of applying the TCN for the underwater image enhancement task.

Sec. 2 provides more implementation details.

Sec. 3 provides more materials of explanation and derivation in the method part.

Sec. 4 provides more results of applying the TCN in other backbones.

Sec. 5 provides more visualization results at the feature level when applying the TCN.

Sec. 6 provides more analysis for the TCN statistics and the affined TCN.

Sec. 7 provides more results and discussions about the toy experiments,

Sec. 8 provides more comparison results about the TCN-Net.

Sec. 9 provides more discussions and results of other TCN formats.

Sec. 10 provides more ablation studies for investigating the TCN.

Sec. 11 provides some discussions in the rebuttal.

Sec. 12 provides more visualization results of various image enhancement tasks.

First of all, we're very sorry there are **some typos** in the main body need to be revised, and we explain and revise them as follows:

(1) **We wrote the normalized feature in Fig.5 (page 7) and Sec. 4.1 (line 206) as $\hat{F}_2$ in the main body, actually, it should be written as $\hat{F}_1$ and the written $\hat{F}_2$ is a typo.**

(2) **The $\sigma_2$ in Eq. 9 (page 4) and Fig.2 (page 5) in the main body lacks a symbol of $\sqrt{}$ in the right of the equation. It should be written as** $\sigma_2 = \frac{1}{\mathrm{HW}} \sum\limits_{i\in[1,\mathrm{H}],j\in[1,\mathrm{W}]} \sqrt{(F_{ij} - \mu_2)^2}$.

## 1  Applying the TCN for Underwater Image Enhancement

In the main body, we have discussed applying the TCN for various image enhancement tasks. Here, to further prove the scalability of the TCN, we apply it for underwater image enhancement. This task aims to correct the color and lightness-shifted appearance of underwater images to clear one, which is an important task in the family of image enhancement tasks.

37th Conference on Neural Information Processing Systems (NeurIPS 2023).

Specifically, we apply the TCN and its variants on the backbone of PUIENet [1], and we perform the experiments on the UIEB dataset [2]. We provide the numeric results in Table 1, and the visualization results in Fig. 18. Both results validate the effectiveness of the TCN.

| Settings | Baseline (PUIENet) | +IN | +TCN | +Affined TCN | +Skip TCN | TCN-Net |
|---|---|---|---|---|---|---|
| PSNR/MS-SSIM | 20.79/0.8029 | 20.13/0.7980 | 21.24/0.8068 | 21.25/0.8075 | 20.86/0.8026 | 21.03/0.7967 |

Table 1: Comparisons over underwater image enhancement.

## 2 More Implementation Details

**Pseudo code of the TCN.** For the implementation of the TCN, we provide the pseudo-code of the original TCN and the affined TCN as follows:

```
def TCN(F):

# F: input with shape [N, C, H, W]
    F = PixelUnshuffle(F)
    # [N, 4C, H/2, W/2]
    F1, F2 = Split(F)
    # F1, F2: [N, 2C, H/2, W/2]
    M2, S2 = CalValue(F2)
    # Calculate values in Eq. 9
    F1 = Normalization(F1,M2,S2)
    # Normalization in Eq. 10
    F2 = F2-F1
    # Eq. 11
    F = Concat(F1, F2)
    # F: [N, 4C, H/2, W/2]

    F = PixelShuffle(F)

    Return F #[N, C, H, W]
```

```
def Affined_TCN(F):

# F: input with shape [N, C, H, W]
    F = PixelUnshuffle(F)
    # [N, 4C, H/2, W/2]
    F1, F2 = Split(F)
    # F1, F2: [N, 2C, H/2, W/2]
    M2, S2 = CalValue(F2)
    # Calculate values in Eq. 9
    M2 = M2+b
    S2 = S2/r
    # b, r are affined parameters
    # Affined operation in Eq. 18
    F1 = Normalization(F1,M2,S2)
    # Normalization in Eq. 19
    F2 = F2-F1
    # Eq. 20
    F = Concat(F1, F2)
    # F: [N, 4C, H/2, W/2]

    F = PixelShuffle(F)

    Return F #[N, C, H, W]
```

Figure 1: **Pseudo-code of the two variants of the proposed TCN.** The left is the *Original TCN* while the right is the *Affined TCN*.

**Implementation details of the TCN-Net.** We implement the TCN in an encoder-decoder architecture, which consists of four scales with a feature channel number of 8. We employ two convolution layers as the basic unit for processing features in one scale. Specifically, it is designed by integrating three Affined TCN operations between the basic unit in the encoder part of the first three scales, and two Skip TCN operations between the first two scales of encoder and decoder. Note that the performance of the TCN-Net could be further improved if other effective blocks can replace the basic unit, and we employ the naive convolution to verify the primary effect of the TCN-Net comes from the TCN.

We train the TCN-Net on a single GTX3090 GPU with a batch size of 4 and total epochs of 1000, the learning rate is set as 8e-4 and decays to half every 200 epochs. The loss function is set as the L1 loss and the training process is end-to-end.

**Implementation details of applying the TCN.** For the Original-TCN and the Affined TCN, we usually apply them before each group of blocks in the network. We apply 3 TCN operations in the first 3 groups. While for the skip-TCN, we apply 2 Skip-TCN operations in the first two levels of the encoder-decoder part in the network.

# 3 More Details of the Method's Material

**Verify the Normalization ability.** In the main body of Eq. 5, we show that $\gamma \approx 1$ satisfies this equation for any number of x. In fact, $\gamma$ can be other numbers, and Eq. 5 could not be satisfied in some cases. In fact, $\gamma$ far away from 1 would change the content, and image enhancement tasks usually recover lightness and color that is content irrelevant.

Besides, we further verify that Eq. 5 could be satisfied for images by a toy experiment. We use the test samples of the MIT-FiveK dataset [3] and augment them using a random number of $\Lambda \in [0.2, 5]$ and $\gamma \in [0.2, 4.0]$. We test the L1 distance between the original and augmented one, as well as their normalized version. The results in Fig. 2 verify the normalization reduces the distance, corresponding to the left part of Fig. 1 in the main body.

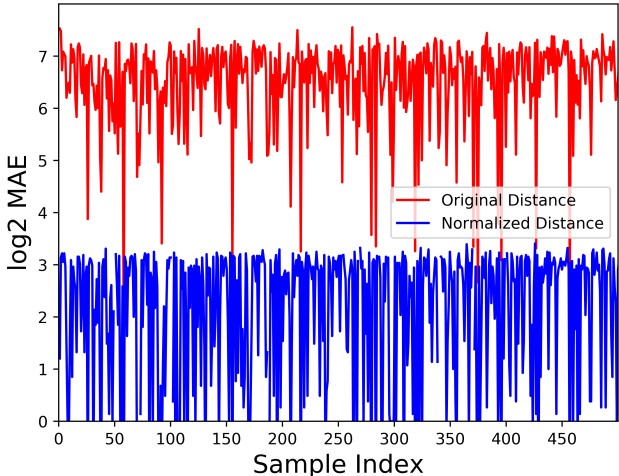

Figure 2: The distance of samples and their augmented one before and after normalization.

**Derivation of the Jacobian Matrix in Eq. 15 of the main body.** We provide the detailed calculation procedure of Eq. 15 in the main body as follows.

Recall Eq. 14 and Eq. 15, we rewrite them here again respectively:

$$
\begin{aligned}
\hat{F}_1 &= (F_1 - M(F_2)) \oslash S(F_2), \\
\hat{F}_2 &= F_2 - F_1 \oslash S(F_2) + M(F_2) \oslash S(F_2),
\end{aligned}
\tag{1}
$$

$$
J_g =
\begin{bmatrix}
\frac{\partial \hat{F}_1}{\partial F_1} & \frac{\partial \hat{F}_1}{\partial F_2} \\[2mm]
\frac{\partial \hat{F}_2}{\partial F_1} & \frac{\partial \hat{F}_2}{\partial F_2}.
\end{bmatrix}
\tag{2}
$$

Then, let $\frac{\partial \hat{F}_1}{\partial F_2} = \triangle$

$$
\frac{\partial \hat{F}_1}{\partial F_1} = \frac{(1-0) \times S(F_2) - 0 \times S(F_2)}{S^2(F_2)} = \frac{1}{S(F_2)}
\tag{3}
$$

$$
\frac{\partial \hat{F}_2}{\partial F_1} = 0 - \frac{1 \times S(F_2) - 0 \times F_1}{S^2(F_2)} + 0 = -\frac{1}{S(F_2)}
\tag{4}
$$

$$
\frac{\partial \hat{F}_2}{\partial F_2} = 1 - \frac{\partial \hat{F}_1}{\partial F_2} = 1 - \triangle
\tag{5}
$$

The Jacobian matrix can be written as:

$$J_g = \begin{bmatrix} \frac{\partial \hat{F}_1}{\partial F_1} & \frac{\partial \hat{F}_1}{\partial F_2} \\ \frac{\partial \hat{F}_2}{\partial F_1} & \frac{\partial \hat{F}_2}{\partial F_2} \end{bmatrix} = \begin{bmatrix} \frac{1}{S(F_2)} & \triangle \\ -\frac{1}{S(F_2)} & 1 - \triangle \end{bmatrix} \tag{6}$$

Employing elementary row and column operations, $J_b$ can be simplified as:

$$J_g = \begin{bmatrix} \frac{1}{S(F_2)} & \triangle \\ -\frac{1}{S(F_2)} & 1 - \triangle \end{bmatrix} \xrightarrow{r_1 + r_2 \to r_2} \begin{bmatrix} \frac{1}{S(F_2)} & \triangle \\ 0 & 1 \end{bmatrix} \tag{7}$$

$$J_g = \begin{bmatrix} \frac{1}{S(F_2)} & \triangle \\ 0 & 1 \end{bmatrix} \xrightarrow{c_1 - c_2 \to c_1} \begin{bmatrix} \frac{1}{S(F_2)} - \triangle & \triangle \\ -1 & 1 \end{bmatrix} \tag{8}$$

$$J_g = \begin{bmatrix} \frac{1}{S(F_2)} - \triangle & \triangle \\ -1 & 1 \end{bmatrix} \xrightarrow{r_1 - r_2 \times \triangle \to r_1} \begin{bmatrix} \frac{1}{S(F_2)} & 0 \\ -1 & 1 \end{bmatrix} \tag{9}$$

We can see the $\mathbf{det}(J_g)$ of Eq. 9 equals to that of Eq. 7, verifying $\mathbf{det}(J_g) \neq 0$, which demonstrate the invertible property of the TCN. We also verify this process follows the invertible process as illustrated in Fig. 3, which is based on the RealNVP's rule [4].

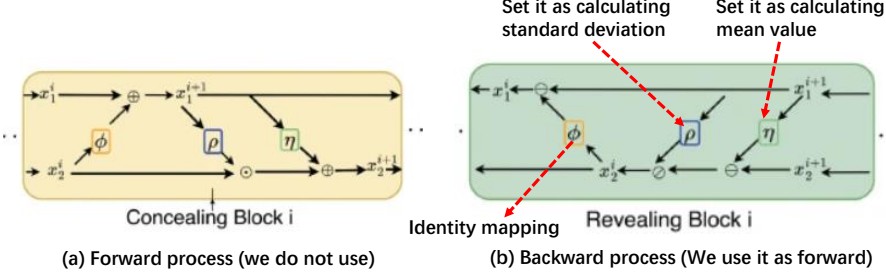

Figure 3: How the TCN is related to the invertible operation.

## 4 Applying the TCN for more backbones

In the main body, we adopt a few networks as the backbone to integrate the TCN. Here, we employ more networks as the backbone to demonstrate the scalability and effectiveness of the TCN, which are described as follows.

**For low-light image enhancement.** We further employ the NAFNet [5] as the backbone, and perform the experiments on the Huawei dataset [6]. The extensive results presented in Table 2 validate the effectiveness of the proposed TCN.

| Settings | Baseline (NAFNet) | +IN | +TCN | +Affined TCN | +Skip TCN |
|---|---|---|---|---|---|
| PSNR/MS-SSIM | 19.51/0.5738 | 18.78/0.5558 | 20.98/0.6210 | 20.96/0.6217 | 20.93/0.6064 |

Table 2: More Comparison over low-light image enhancement on the Huawei dataset.

**For exposure correction.** We further employ the LCDPNet [7] as the backbone for exposure correction. We perform the experiments in the SICE dataset [8], and the experimental results in Table 3 demonstrate the effectiveness of the proposed TCN.

| Settings | Baseline (LCDPNet) | +IN | +TCN | +Affined TCN | +Skip TCN |
|---|---|---|---|---|---|
| PSNR/MS-SSIM | 20.46/0.6941 | 20.60/0.6880 | 20.89/0.7026 | 20.95/0.7047 | 20.80/0.6992 |

Table 3: More Comparison for exposure correction on the SICE dataset.

**For SDR2HDR translation.** Moreover, we adopt the DRBN [9] as the backbone for SDR2HDR translation. The experimental results performed on the sritm dataset [10] the HDRTV dataset [11] in Table 4 demonstrate the effectiveness of the proposed method.

**For image dehazing.** We also employ the AODNet [12] as the backbone to perform the experiments in Table 5. The results on the RESIDE dataset [13] validate the TCN's effectiveness.

| Settings | Baseline (DRBN) | +IN | +TCN | +Affined TCN | +Skip TCN |
|---|---|---|---|---|---|
| sritm dataset | 32.52/0.9435 | 32.39/0.9420 | 32.88/0.9471 | 32.94/0.9486 | 32.77/0.9462 |
| HDRTV dataset | 35.48/0.9585 | 35.11/0.9532 | 35.61/0.9597 | 35.53/0.9578 | 35.76/0.9604 |

Table 4: More Comparison over SDR2HDR translation in terms of PSNR/SSIM.

| Settings | Baseline (AODNet) | +IN | +TCN | +Affined TCN | +Skip TCN |
|---|---|---|---|---|---|
| Indoor | 19.69/0.8289 | 19.08/0.8127 | 19.74/0.8300 | 19.69/0.8292 | None |
| Outdoor | 23.62/0.9214 | 24.17/0.9215 | 23.78/0.9283 | 23.77/0.9248 | None |

Table 5: More Comparison over image dehazing in terms of PSNR/SSIM.

# 5 More Visualization Results of Features

We provide the visualization results of features in the TCN based on the SID as an illustration, which is trained on the MIT-FiveK dataset. Specifically, as depicted in Fig. 4, the TCN can transform the feature to a more structured one, demonstrating its ability to extract consistent structure representation.

Moreover, as shown in Fig. 5, the TCN helps stabilize the gradient of the SID in deep layers, which reflects the TCN could help improve the optimization effect, which has been validated in previous normalization studies [14, 15].

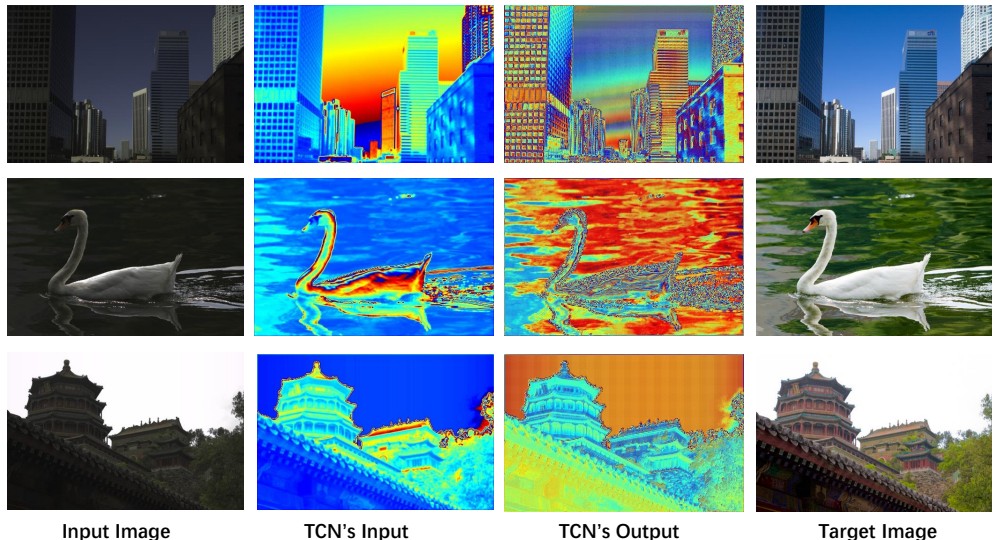

| Input Image | TCN's Input | TCN's Output | Target Image |

Figure 4: Visualization results of the TCN features on SID baseline trained on the MIT-FiveK dataset.

# 6 More Analysis for Affined TCN and TCN's Statistic.

**Validate the transition-constant of the affined TCN.** We further validate the affined TCN satisfies the invertible constraint that transits information constantly as follows.

Let's revisit Eq. 13 in the main body, which is written as:

$$\hat{F}_1 = (F_1 - M(F_2)) \oslash S(F_2),$$
$$\hat{F}_2 = F_2 - \hat{F}_1, \tag{10}$$

Following the above format, the affined TCN in Eq. 20 of main body can be expressed as:

$$\hat{F}_1 = (F_1 - M(F_2) - \beta) \oslash S(F_2) \odot \gamma,$$
$$\hat{F}_2 = F_2 - \hat{F}_1, \tag{11}$$

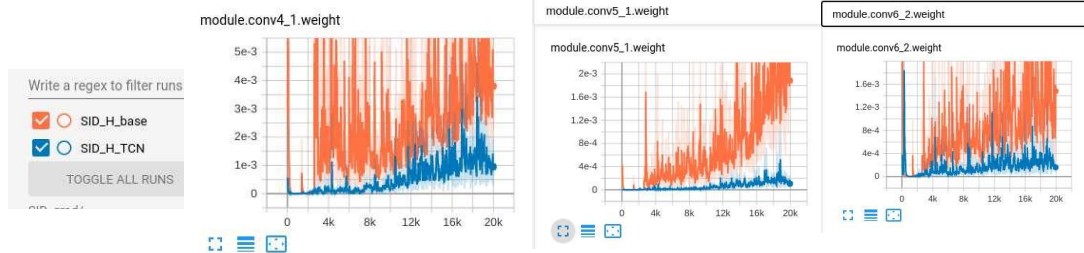

Figure 5: The parameter fluctuation in different layers, the TCN stabilizes the weight changes.

where $\odot$ denotes the element multiplication. Then, it can be converted as:

$$\hat{F}_1 = (F_1 - M(F_2) - \beta) \oslash S(\frac{F_2}{\gamma}),$$
$$\hat{F}_2 = F_2 - F_1 \oslash S(\frac{F_2}{\gamma}) + M(F_2) \oslash S(\frac{F_2}{\gamma}) + \beta \oslash S(\frac{F_2}{\gamma}), \tag{12}$$

We calculate the Jacobian matrix of Eq. 12 as:

$$J_g = \begin{bmatrix} \frac{\partial \hat{F}_1}{\partial F_1} & \frac{\partial \hat{F}_1}{\partial F_2} \\ \frac{\partial \hat{F}_2}{\partial F_1} & \frac{\partial \hat{F}_2}{\partial F_2} \end{bmatrix} = \begin{bmatrix} \frac{1}{\partial S(\frac{F_2}{\gamma})} & 0 \\ -1 & 1 \end{bmatrix} \tag{13}$$

The derivation process is the same as those in Sec.3 in the supplementary material and we omit here. Upon the $\mathbf{det}(J_g) \neq 0$, it indicates that $J_g$ is full rank, verifying the invertible property of Affined TCN and further the transition-constant ability.

**Validate $\mu_2 \approx \mu_1$ and $\sigma_2 \approx \sigma_1$ in the TCN.** We then further validate the $\mu_2$ and $\sigma_2$ satisfy the $\mu_2 \approx \mu_1$ and $\sigma_2 \approx \sigma_1$ by the toy experiment in Sec. 4.1 in the main body. Specifically, we replace the $\mu_2$ and $\sigma_2$ with $\mu_1$ and $\sigma_1$ in Eq. 9 in the main body respectively to demonstrate the reconstruction results are very similar to the original reconstruction results. We present the experimental settings in Fig. 6, and the results are shown in Table 6.

As can be seen, setting $\mu_2, \sigma_2$ to either $\mu_1, \sigma_1$ would lead to nearly the same reconstructed results, demonstrating they are correspondingly the same. Moreover, these statistics also correspond to those of the original feature $F$ (Setting (a)). When the statistics have changed, the reconstructed results would have lower similarity to original one (Setting (b)). All results can validate $\mu_2 \approx \mu_1, \sigma_2 \approx \sigma_1$.

| Settings | Set $\mu_2$ as $\mu_1$ | Set $\sigma_2$ as $\sigma_1$ | Set $\mu_2$ ,$\sigma_2$ as $\mu_1$,$\sigma_1$ | Set $\mu_2$ ,$\sigma_2$ as (a) | Set $\mu_2$ ,$\sigma_2$ as (b) |
|---|---|---|---|---|---|
| PSNR (dB) | 61.95 | 59.12 | 59.11 | 62.15 | 45.21 |

Table 6: Reconstruction results by replacing $\mu_2, \sigma_2$ in Eq. 9 with other statistics. (a) denotes the statistics of the original feature $F$, (b) denotes the statistics of the half-part channels in $F$.

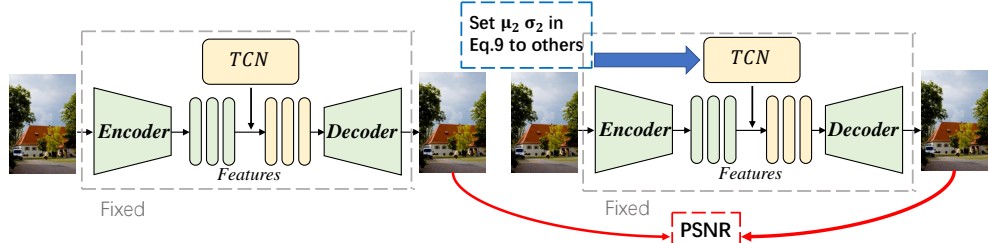

Figure 6: The experiment illustration corresponding to Sec. 6 and Table 6 in the supplementary.

# 7 More Materials of Toy Experiments

We provide more results of the toy experiments here. We provide the reconstruction PSNR of each test sample on the MIT-FiveK dataset for toy experiment in Fig. 7 (a), which also validates the TCN has better transition-constant ability than the IN.

Moreover, besides the features provided in the main body, we further test the average distance between the underexposure and overexposure pairs feature in the TCN. As illustrated in Fig. 7 (b), the TCN has effectively reduced their distances, demonstrating the normalization ability of the TCN for capturing lightness-consistence representations. We further present the feature maps in Fig. 8, which also validates the above conclusion intuitively.

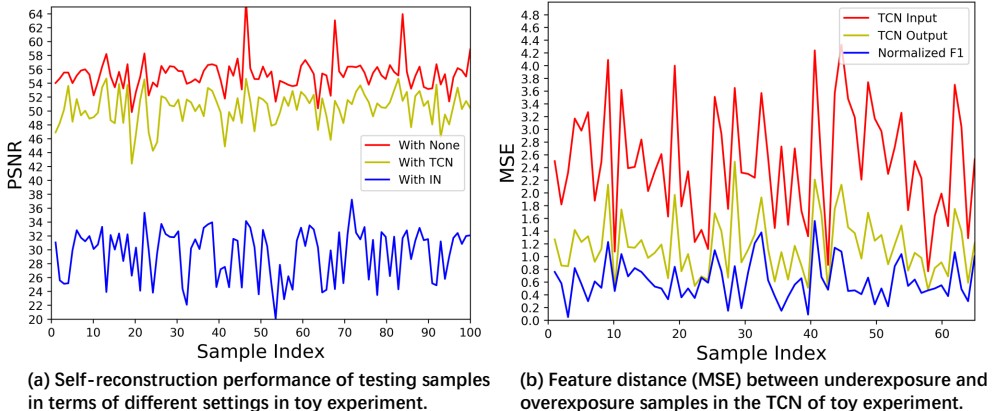

(a) Self-reconstruction performance of testing samples in terms of different settings in toy experiment.

(b) Feature distance (MSE) between underexposure and overexposure samples in the TCN of toy experiment.

Figure 7: Toy experiment results on sample levels. The left is the self-reconstruction similarity of testing samples, and right is the similarity of underexposure and overexposure pair features in TCN.

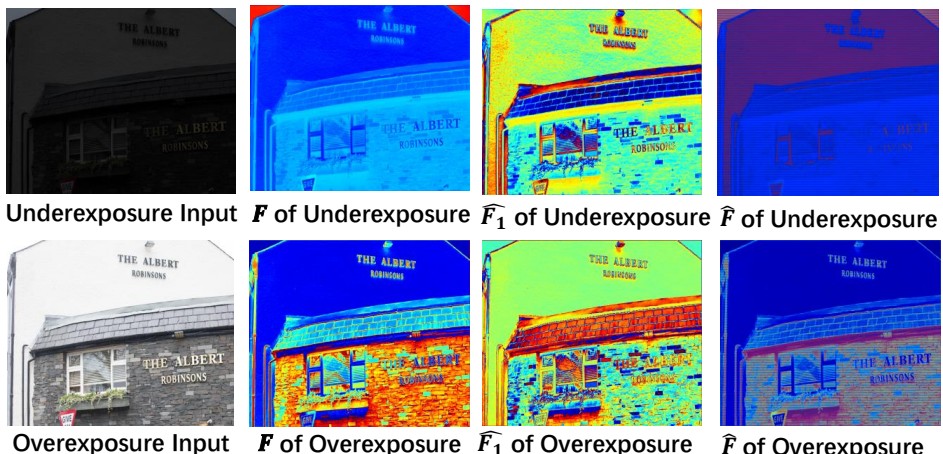

Figure 8: Feature visualization of toy experiment. We can see the normalized features (3rd row) and the output features of the TCN (last row) of underexposure and overexposure become more similar.

# 8 More results about the TCN-Net

We provide more quantitative results of the TCN-Net and other comparison methods on the low-light enhancement and exposure correction. The results are presented in Fig. 9, which demonstrates the TCN-Net achieves an elegant balance between performance and efficiency.

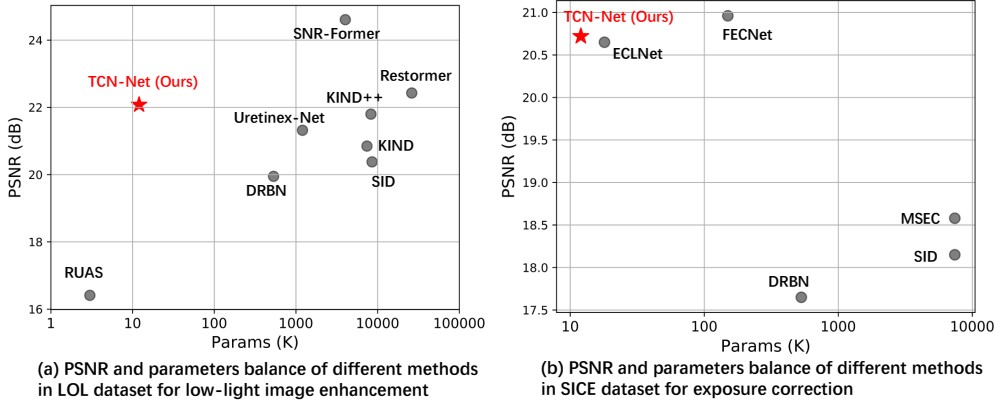

(a) PSNR and parameters balance of different methods in LOL dataset for low-light image enhancement

(b) PSNR and parameters balance of different methods in SICE dataset for exposure correction

Figure 9: Balance trade-off between PSNR and parameters for different methods.

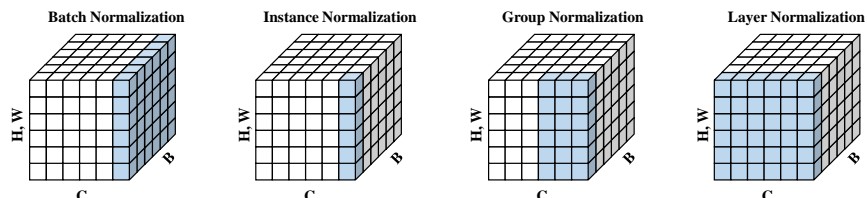

Figure 10: The normalization family of different operations, including BN, IN, GN and LN.

## 9 More Materials of Other TCN Formats

In Sec.3.3 of the main body, besides IN, we have discussed the calculation of $\mu_2$ and $\sigma_2$ can be other formats, resulting in other TCN formats based on LN, BN, or GN. To depict them besides Eq. 2 in the main body, we illustrate them in Fig. 10.

Then, we conduct experiments by extending the TCN of these formats on the MIT-FiveK dataset [3], which is based on the SID backbone. We present the results in Table 7, proving the effectiveness of the other TCN formats for image enhancement. For GN, its group size is set as 4, and each group has the same number of channels. In the future, we will investigate more usage of them in other tasks.

| Settings | Baseline (DRBN) | +TCN(IN) | +TCN(LN) | +TCN(GN) | +TCN(BN) |
|----------|-----------------|----------|----------|----------|----------|
| PSNR/MS-SSIM | 22.11/0.8684 | 23.98/0.8851 | 23.91/0.8846 | 24.02/0.8859 | 22.67/0.8733 |
| Settings | Baseline (SID) | +TCN(IN) | +TCN(LN) | +TCN(GN) | +TCN(BN) |
| PSNR/MS-SSIM | 21.49/0.8425 | 23.11/0.8581 | 23.36/0.8607 | 23.42/0.8595 | 21.58/0.8510 |

Table 7: Investigating more TCN formats of other normalization operations on the MIT-FiveK dataset.

**More discussions.** (1) Since there exist various invertible frameworks, the TCN can be constructed in other formats if the invertible rules can be satisfied such as the illustration in Fig. 3. (2) The subsampling manner for dividing the two streams can be other manners which satisfy they have nearly the same statistics ($\mu_2 \approx \mu_1$ and $\sigma_2 \approx \sigma_1$). After all, the core of our method is its design concept rather than its implementation formats, **which could inspire more work toward designing transition-constant operations for image enhancement or image restoration tasks.**

## 10 Ablations for Investigating the TCN.

We further investigate the core design of the TCN to validate the reasonableness of its design. We conduct the experiments on the SID baseline on the MIT-FiveK dataset [3]. We present the two

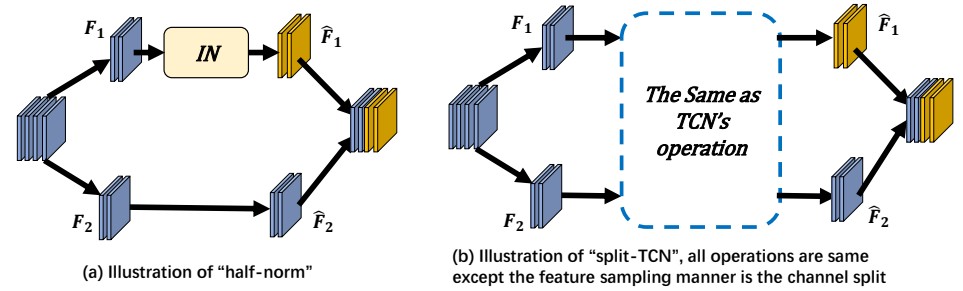

(a) Illustration of "half-norm"

(b) Illustration of "split-TCN", all operations are same except the feature sampling manner is the channel split

Figure 11: The ablation settings for comparison.

| Settings | Baseline (DRBN) | +IN | +half-norm | +split-TCN | +TCN |
|---|---|---|---|---|---|
| PSNR/MS-SSIM | 22.11/0.8684 | 22.93/0.8727 | 23.80/0.8839 | 23.76/0.8812 | 23.98/0.8851 |

Table 8: Alations for investigating the TCN.

configurations as shown in Fig. 11. The left one denotes that only half of all channels are instance normalized and we name it "half-norm", which only satisfies the normalization ability but has limited transition-constant ability without invertible constraint. The right one denotes we replace the sub-sampling operation of unshuffle with channel split, and we name it "split-TCN". It only satisfies the transition-constant ability but has limited normalization ability due to the inconsistent statistics of the two streams ($\mu_2 \approx \mu_1$ and $\sigma_2 \approx \sigma_1$ can not satisfy). The results in Table 8 validate that the TCN outperforms these two settings and proves its effectiveness.

Moreover, referring to the setting of the toy experiment, we also validate that the "half-norm" in Sec. 10 can not self-reconstruct well as the TCN as shown in Fig. 12 (HIN is "half-norm" here), which proves its limited transition-constant ability for reconstructing itself.

## 11 Some Discussions in the Rebuttal

**About the relationship of transition-constant ability and image enhancement.** First, for architectures such as CNN and transformer, they convey information with the driven of reconstruction losses, thus the whole part of them is nearly transition-constant. However, Instance normalization (IN) discards information [16] instead of learning to filter the information like CNN or transformer, making the sub-sequential parts cannot restore the information. This is how IN's weakness is different

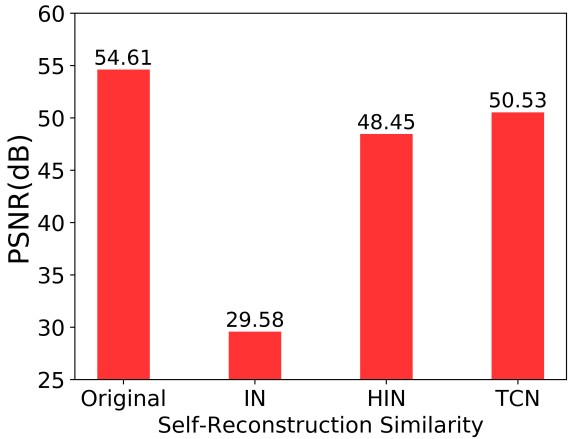

Figure 12: The self-reconstruction results when inserting different operations in the toy experiments. We additionally include the reconstruction result of "half-norm (HIN)" here.

from other architectures. Second, due to the IN's weakness, and its advantage is normalizing different lightness and thus is useful for image enhancement tasks, we improve the IN with the transition-constant ability that enables IN can be practically used for image enhancement. Note that this ability is equipped for IN instead of other parts. Finally, DSN [17] shows the transition-constant ability of networks can be developed for image enhancement, while this work doesn't explore IN's property.

**About applying TCN in small features or features with odd number resolution.** For the first question, we implement TCN on the feature with large-size features, while small-size features contain much less information than the large size, therefore applying the TCN in such information-reduced features is not necessary. As for the latter question, in experiments, most images' resolution can be divided by 2, and we usually apply TCN in the large features whose sizes can be divided by 2. When applying TCN on features with odd number resolution, we need to resize the feature to the nearest even resolution preliminary.

**Discussion with BN.** Firstly, BN is a kind of normalization that has the best capability of keeping information representation property [18]. Therefore, BN can transit almost all information and thus can be employed for some low-level vision tasks directly. Differently, other normalization formats especially IN would destroy information representation according to [1], which is different from BN. Secondly, for image enhancement, IN is useful for capturing lightness-consistent representation according to Fig.1, it depicts IN has the potential for image enhancement once with transition constant property. Meanwhile, BN is not very effective for capturing low-level statistics and thus is not suitable for image enhancement as shown in supplementary (Table 7). This is why our work is different from the mentioned two works. We will further clarify this point in the updated version.

**Discussion with LN.** In fact, LayerNorm(LN) plays the role of balance activation function in the Transformer, which is different from what we explore for image enhancement. Besides, LN has a weaker activation ability to the feature than IN, thus improving IN with TCN formats is more meaningful to extend it for normalizing lightness while conveying the information constantly

## 12 More Qualitative Results

Due to the page limit of the main body, we provide more visualization results here. We respectively present the results of low-light image enhancement (Fig. 14 and Fig. 13), exposure correction (Fig. 15), SDR2HDR translation (Fig. 16), image dehazing (Fig. 17) and underwater image enhancement 18 as follows. As can be seen, our TCN can help enhance more correct lightness and color, or reduce the structure artifacts.

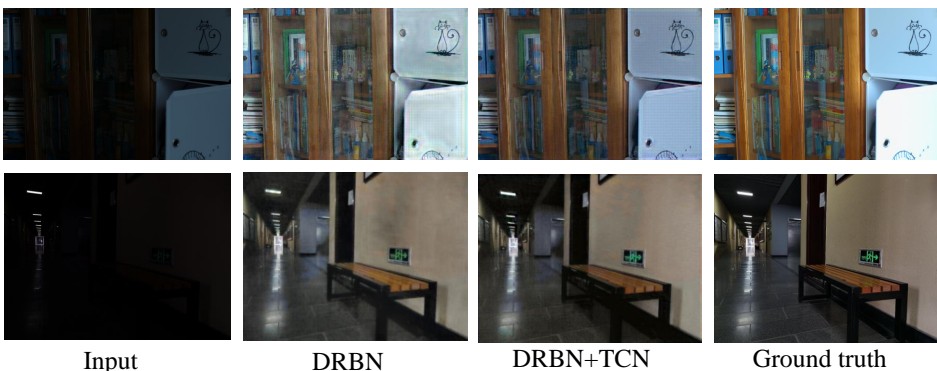

| Input | DRBN | DRBN+TCN | Ground truth |

Figure 13: Visual results of low-light image enhancement on LOL (up) and Huawei (down) datasets.

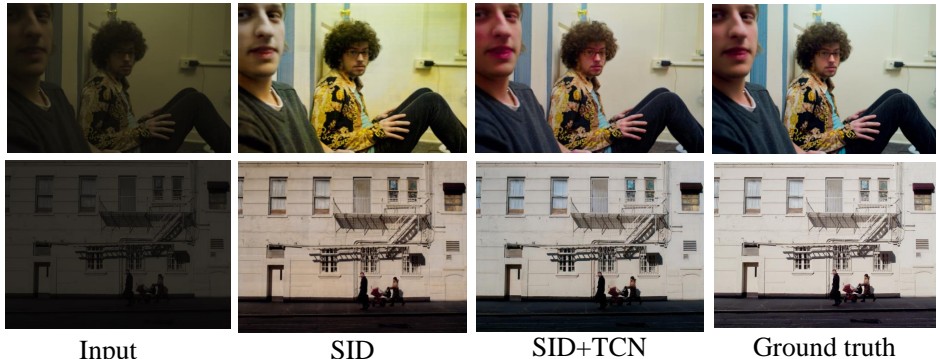

| Input | SID | SID+TCN | Ground truth |

Figure 14: Visual results of low-light image enhancement on MIT-FiveK dataset.

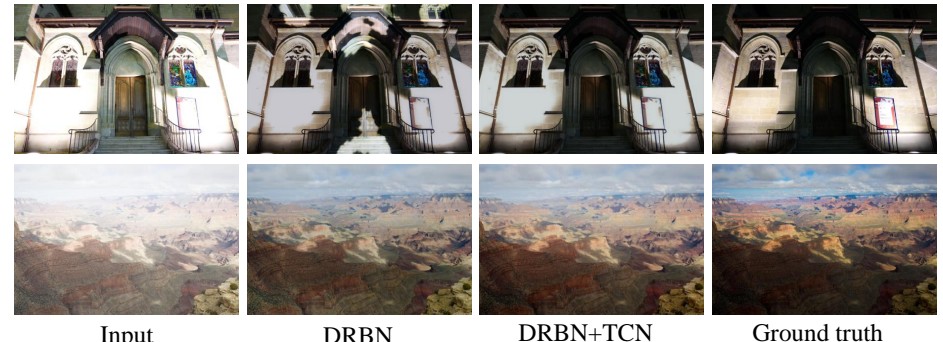

| Input | DRBN | DRBN+TCN | Ground truth |

Figure 15: Visual results of exposure correction on SICE (up) and MSEC (down) datasets.

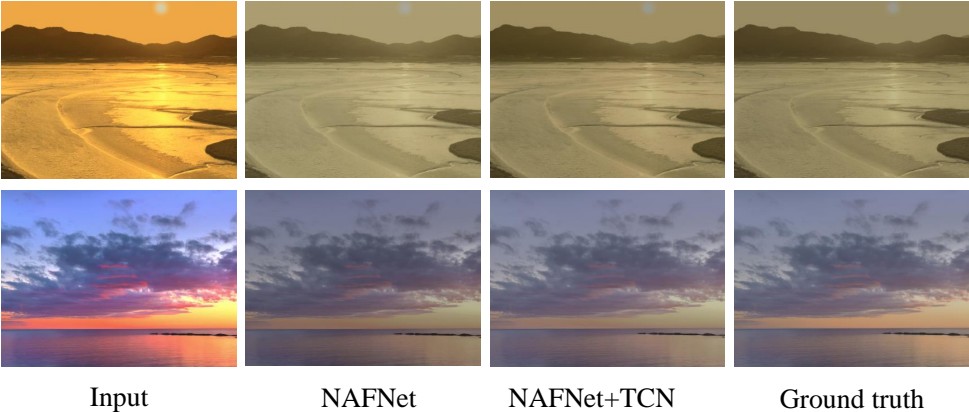

| Input | NAFNet | NAFNet+TCN | Ground truth |

Figure 16: Visual results of SDR2HDR translation on HDRTV dataset (up) and sritm dataset (down).

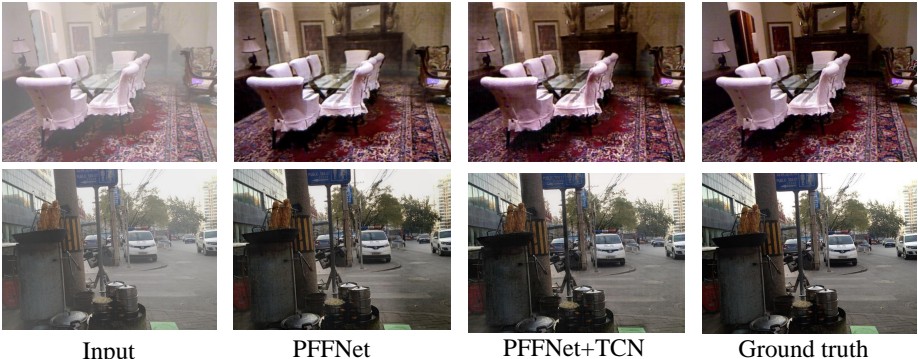

| Input | PFFNet | PFFNet+TCN | Ground truth |

Figure 17: Visual results of image dehazing on SOTS indoor dataset (up) and outdoor dataset (down).

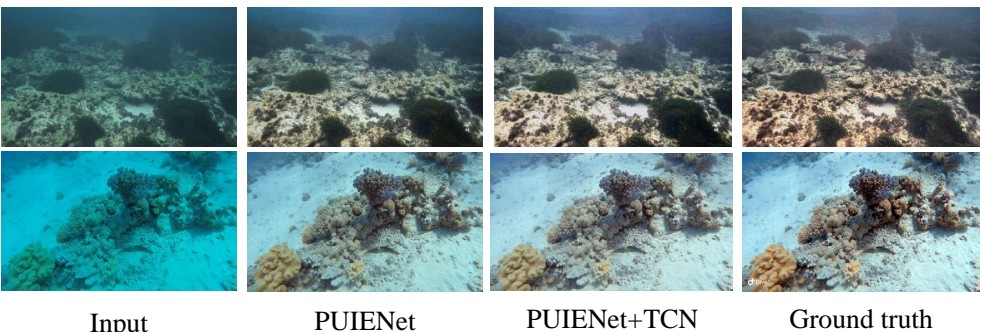

| Input | PUIENet | PUIENet+TCN | Ground truth |

Figure 18: Visual results of underwater image enhancement on UIEB dataset.