# OpenReview forum: "Transition-constant Normalization for Image Enhancement"
_NeurIPS.cc/2023/Conference — NeurIPS 2023 spotlight_

### Official Review · Reviewer_yudq · 2023-07-03

**Soundness:** 4 excellent
**Presentation:** 3 good
**Contribution:** 3 good
**Rating:** 8
**Confidence:** 5

**Summary:**

This paper presents a novel Transition-Constant Normalization (TCN) for various image enhancement tasks. It has several formats and is plug-and-play for existing image enhancement networks. In the experimental part, proposed method show the extensibility and superiority of the proposed method on both quantitative and qualitative comparisons.

**Strengths:**

There are several strengths here:
1. Overall, the motivation is clear presented and seems to be reasonable. The customized design of normalization technique could be a meaningful point.
2. The method design is reasonable and comprises various formats, the inner key design is to ensure transition-constant, making the concrete format of the method is flexible when applying in specific tasks.
3. Experiments verify the effectiveness of the proposed method, and exhibit good extensibility in various scenes.


**Weaknesses:**

There are several weakness here:
1. Although the work is based on designing transition-constant normalization, previous methods have employed normalization in low-level vision tasks. For example, Dncnn incorporate batch normalization and RepSR also employ batch normalization and show it can improve performance. These raised cases make the statement of the proposed method is a little puzzled.
2. The main figure of describing the method is not straightforward to show its transition-constant, I need to read very carefully to figure out its mechanism.
3. The baseline that incorporates the TCN is old in some tasks such as low-light image enhancement tasks, this could bring doubt whether the TCN is compatible with more recent architectures.


**Questions:**

I have several concerns here:
1. Can authors explain why the existing method is not suitable to use normalization although there exists Dncnn and RepSR.
2. Can authors provide more recent baselines to demonstrate the effectiveness of the TCN?


**Limitations:**

Yes, authors have discussed the limitations at the end of the main body. I am also curious about whether the proposed method could be applied in other scenes.

---

> ### Author Rebuttal · Authors · 2023-08-09
>
> Thanks for your time and efforts and here we present the rebuttal.
>
> **1. Discussion with BN.**
>
>  Firstly, BN is a kind of normalization that has the best capability of keeping information representation property [1]. Therefore, BN can transit almost all information and thus can be employed for some low-level vision tasks directly. Differently, other normalization formats especially IN would destroy information representation according to [1], which is different from BN. Secondly, for image enhancement, IN is useful for capturing lightness-consistent representation according to Fig.1, it depicts IN has potential for image enhancement once with transition constant property. Meanwhile, BN is not very effective for capturing low-level statistics and thus is not suitable for image enhancement as shown in supplementary (Table 7). This is what our work is different from the mentioned two works. We will further clarify this point in the updated version.
>
>  [1] Antoine et al. Proxy-Normalizing Activations to Match Batch   Normalization while Removing Batch Dependence. NeurIPS 2021.
>
> **2. The main figure is not straightforward.**
>
> Thanks for pointing out this point. In fact, we present the main figure to demonstrate its procedure, and its relevance to the transition-constant property has been proved in supplementary(Fig. 3). We will clarify the relation of the main figure to Fig. 3 in supplementary in the updated version to make it clear.
>
>
> **3. More results applying TCN in recent baselines?**
>
> Thanks for your suggestion. We have already presented results on the recent method NAFNet(ECCV 2022) in the supplementary. Due to limited time and space, here we present the results of Restormer (CVPR 2022) and Bread (IJCV 2023) on the LOL dataset here. We will add these results in the revision. Note that we also propose a lightweight TCN-Net constructed by the TCN and it achieves a good trade-off between effectiveness and efficacy.
>
> | Model| Methods|PSNR|SSIM|
> |----|----|----|----|
> | Restormer|Original|22.24|0.828|
> |        |+IN|22.06|0.824|
> |        |+TCN|22.47|0.831|
> | Bread|Original|22.96|0.838|
> ||+IN|22.85|0.836|
> ||+TCN|23.26|0.841|
>
>
> **4. Extensive ability.**
>
> Since TCN is developed for the style-based task of image enhancement, we think that the TCN can also be extended to other tasks about style processing, such as domain adaptation and multi-sensor image fusion. The extensive experiments on pan-sharpening (fusing two style images) and medical segmentation (there’s a domain between training and testing) can illustrate it.

---

> > ### Comment · Reviewer_yudq · 2023-08-17
> > **Response to Authors' Comments**
> >
> > Thanks for your response. After the rebuttal, my concerns have been addressed, and the authors are suggested to add their rebuttal comments to the paper. Overall, this paper is interesting in proposing several extensive formats, and experiments well depict the effectiveness.

---

### Official Review · Reviewer_5FQQ · 2023-07-04

**Soundness:** 4 excellent
**Presentation:** 3 good
**Contribution:** 3 good
**Rating:** 8
**Confidence:** 5

**Summary:**

The authors introduce a series of normalization formats by incorporating the transition-constant constraint design. The motivation behind the work is presented, extensive experiments across multiple image enhancement tasks are carried out.

**Strengths:**

i. The idea is interesting. With the introducing of invertible mechanism, normalization operation achieves constant information transition. Its property is satisfied with the requirement of image enhancement while reducing the negative effect of blocking information conveying.
ii. The authors presented the method design from motivation analysis and concrete implementation. They have conducted experiments across diverse scenarios, showing the method's efficacy and its wide range of applications.
iii The paper is well organized and easy to be understood. Notably, some information is important when reviewing supplementary materials.


**Weaknesses:**

i. Authors presented the motivation，but it seems that achieving information transition can be accomplished through alternative manners. For example，adding a residual pass from the pre-normalized features could also serve this purpose. Nevertheless, authors have not provided a thorough discussion.
ii. Meanwhile，transformer that contains layer norm is a specific case of above discussed problems. Authors should also discuss it.
iii. In Eq.4，authors assume the lightness changes rule. Nevertheless, quantities of image enhancement tasks may not follow this rule. Indeed, normalization is able to normalize different representation。
iv. Some conducted baselines are too simple to prove the effectiveness of the introduced technique, such as SID.


**Questions:**

Please see [Weaknesses], authors need to answer the above problems.

**Limitations:**

The paper comprises the limitation part and addresses points from different aspects.

---

> ### Author Rebuttal · Authors · 2023-08-09
>
> Thanks for your time and efforts and here we present the rebuttal.
>
>
> **1. About adding a residual for normalization.**
>
>  In fact, adding a residual path is not a reasonable manner: Firstly, it adds the original information to the normalized one directly, which does not consider their relationship for conveying information and thus is not transition inconstant; Secondly, adding the original information and the normalized one would blend their information, leading to the network to learn to reduce the blend effect that integrates them properly, thus is not effective.
>
>
> **2. About discussion the LayerNorm.**
>
> There are several reasons that the LayerNorm(LN) is not a good format as the TCN. First, IN has the best ability to activate the features [1] that are normalized to lightness-invariant (thus style transfer often employs IN[2] instead of LN), and features are lower activated to LN.  Second, LN in the transformer is for stabilizing the training procedure [3], its role is different from what we explore for image enhancement and the residual connection is not designed for our goal as depicted in the above answer. Third, experiments of NAFNet with LN in supplementary (Table 2) show that the TCN can improve its performance, showing our TCN is compatible with the LN for improving image enhancement.  Moreover, we also present an experiment on the MIT-FiveK dataset by applying LN on the DRBN here, depicting applying LN directly has limited performance compared with TCN.
>
>  | Model | Methods |PSNR | SSIM |
> |----|----|----|----|
> | DRBN| Original | 22.11 | 0.868  |
> |        | +LN| 22.84  | 0.874 |
> |        | +TCN | 23.98  | 0.885 |
>
> [1] Antoine et al. Proxy-Normalizing Activations to Match Batch Normalization while Removing Batch Dependence. NeurIPS 2021.
>
> [2] Vedaldi et al. Instance Normalization: The Missing Ingredient for Fast Stylization. arXiv:1607.08022
>
> [3] Yu et al. MetaFormer Is Actually What You Need for Vision. CVPR 2022.
>
>
> **3. About depicting the normalization ability.**
>
> We present Eq. 4 to further demonstrate that normalization has the ability to normalize various lightness besides Fig. 1. Other image enhancement formats are complicated and are not easy to present. After all, since the enhancement task aims to change the overall style to visually pleasant, normalization is suitable for normalizing styles as depicted in style transfer works.
>
>
> **4. About Experiments.**
>
> Thanks for your suggestion. As depicted in supplementary (Table 2), we have performed experiments in some recent and complicated architectures such as NAFNet and PFFNet. Here, we present the results of another two recent backbones Restormer (CVPR 2022) and Bread (IJCV 2023) on the LOL dataset in low-light image enhancement to demonstrate the effectiveness. It is important to emphasize that the core of our method is to improve the performance of existing works rather than provide a SOTA, and the lightweight TCN-Net is also useful. We will add more discussions about it in the revision.
>
> | Model | Methods |PSNR | SSIM |
> |----|----|----|----|
> | Restormer| Original | 22.24 | 0.828|
> |        | +IN| 22.06  | 0.824  |
> |        | +TCN | 22.47| 0.831|
> | Bread|Original|22.96|0.838|
> ||+IN|22.85|0.836|
> ||+TCN|23.26|0.841|

---

> > ### Comment · Reviewer_5FQQ · 2023-08-15
> > **Response to Rebuttal**
> >
> > Thanks for the author's thoughtful reply. The rebuttal addressed my concerns well. I was originally positive at the paper. When I checked other reviews and the rebuttal, I decided to raise my rating. Besides, it is better to add these analyses in the rebuttal to the released version.

---

### Official Review · Reviewer_uwKS · 2023-07-05

**Soundness:** 3 good
**Presentation:** 2 fair
**Contribution:** 3 good
**Rating:** 5
**Confidence:** 4

**Summary:**

This paper proposes an invertible feature normalization module based on the design of coupling layers in flow-based models [1]. The authors experimentally evaluate the proposed model on various image enhancement tasks such as low-light image enhancement, HDR, image dehazing, as well as medical segmentation and guided image super-resolution as downstream tasks. The results show improvements over their baseline methods.

[1] Density estimation using Real NVP, ICLR 2017.

**Strengths:**

The paper covers a range of tasks in the experiments. The module design is simple and achieves comparable speed to previous feature normalization methods.

**Weaknesses:**

(1) If I understand correctly, the necessary statement and property are already provided by [1] in its Section 3.7. Then the contribution of this paper is limited in the application of this formulation. Meanwhile, why does the author emphasize the importance of the so-called "transition constant" in image reconstruction? After all, the current normalization methods (such as BN, IN, LN, GN, etc.) work well in high-level vision tasks. To my knowledge, IN may indeed not be suitable for image reconstruction tasks (to some extent), but BN and LN used in many low-level vision tasks are appropriate. However, the author only compared IN in their toy experiments, which I find inadequate.

[1] Density estimation using Real NVP, ICLR 2017.

(2) The comparison methods are severely lacking, e.g., DRBL [1], KinD++ [2], Bread [3] for low-light image enhancement and FFA-Net [4], MAXIM [5], UDN [6] and C^2PNet [7] for image dehazing. All the other comparisons have the same problem, and I just name two of them. I do not think comparing two or fewer methods is enough to illustrate the validity of a method.

[1] From fidelity to perceptual quality: A semi-supervised approach for low-light image enhancement. In CVPR, 2020.

[2] Beyond brightening low-light images. IJCV, 2021.

[3] Low-light image enhancement via breaking down the darkness. IJCV, 2023.

[4] FFA-Net: Feature fusion attention network for single image dehazing. In AAAI, 2020.

[5] Maxim: Multi-axis mlp for image processing. In CVPR 2022.

[6] Uncertainty-driven dehazing network. In AAAI, 2022.

[7] Curricular Contrastive Regularization for Physics-aware Single Image Dehazing. In CVPR 2023.

(3) It is unclear why the author chose medical image segmentation and pan-sharpening tasks for downstream applications instead of testing on datasets with low-light or adverse weather conditions, such as DARKFACE [1], EXDARK [2], and RTTS [3].

[1] Advancing Image Understanding in Poor Visibility Environments: A Collective Benchmark Study. TIP, 2020.

[2] Getting to know low-light images with the Exclusively Dark dataset. CVIU, 2019.

[3] Benchmarking single-image dehazing and beyond. TIP, 2019.


(4) Please correct the format in lines 143-156, as it appears to be confusing.

To be concluded, the main drawbacks of this paper are the lack of strong motivation and insufficient experiments, which fail to provide sufficient persuasion. The consistency in the style of the figures and the layout of the paper also need improvement.

**Questions:**

Please see the Weaknesses.

**Limitations:**

This paper has no apparent negative societal impact.

---

> ### Author Rebuttal · Authors · 2023-08-09
>
> Thanks for your time and efforts and here we present the rebuttal.
>
> **1. About motivation and contribution.**
>
> The meaningful of our work: (1) Since IN can normalize different lightness effectively, we have shown that equipping IN as TCN with transition constant ability is useful for image enhancement. (2) Based on it,  the formula statement is a theoretical demonstration to show that TCN satisfies transition-constant, which is necessary. Meanwhile, the toy experiment also validates that the TCN indeed satisfies this property experimentally. Both are necessary to depict our method. (3) Therefore,  our work is meaningful that brings a new view to improve IN as TCN for image enhancement, and is useful for constructing lightweight models.
>
> The importance of transition-constant ability: (1) Experimentally, IN would lead to information loss [1] and huge representation ability changes[2][3], therefore, what we address for “transition constant” is to equip IN with this property for image enhancement, and the toy experiment of IN depict it. (2) Previous work[4] shows that the invertible transition-constant also benefits image enhancement. Here, we focus on equipping normalization (especially IN) and making it more suitable for image enhancement. We will further clarify the above points.
>
> About IN and other normalizations: (1) IN is suitable: IN has the strongest ability to change information representation than other normalizations [2], and is often employed for style-transfer tasks while other normalization seldom does it, thus improving IN is more suitable for image enhancement (related to process style).  (2) BN and LN are not the focus of our paper: BN and LN less affect information representation ability according to [2], therefore, previous work can adopt them in some low-level backbones. However,  IN is more suitable than BN and LN for style-based tasks (such as style transfer), and although IN can not be used for image enhancement (which is related to enhancing style) directly,  what we do in this paper is to develop IN with TCN to be suitable for image enhancement. Moreover, In supplementary (Table 7), we have also shown that the IN-based approach has better performance for image enhancement than the BN-based approach. (3) More about LN:  We have provided results of applying TCN on the NAFNet (it has LN) in supplementary (Table 2), depicting that TCN can further improve performance with LN. As suggested, we also supplement the toy experiments about other normalization (BN can transit information refers to [2]) to make it clearer, we will add them and more discussions in revision.
> Methods |PSNR|
> |----|----|
> |LN|48.67|
> |TCN(LN)|53.42|
> |GN|41.53|
> |TCN(GN)|51.74|
> |None|54.61|
>
> **2. About experiments.**
>
> Experiments number: (1) We have conducted 3 baselines for most tasks including the experiments in the supplementary. (2) We have conducted 5 image enhancement tasks and 2 extensive tasks, which are enough. (3) For TCN-Net, we also provide more comparison results in supplementary (Fig. 9) with 8 methods including KIND++, demonstrating its good trade-off between efficacy and effectiveness.
>
> We also need to illustrate that the contribution of our method is to bring a new perspective that enables normalization technique (especially IN) can be useful for image enhancement. Therefore, experiments among three backbones can illustrate it.
> As suggested, we also provide more results with the suggested methods. Since DRBL (we denote as DRBN) has been provided in the main part and there’s no code of UDN, we provide the results of other methods on the LOL dataset and Indoor dataset here.
> |Model|Methods|PSNR|SSIM|
> |----|----|----|----|
> |KIND++| Original|21.30|0.823|
> ||+IN|22.10|0.824|
> ||+TCN|23.03|0.829|
> | Bread|Original|22.96|0.838|
> ||+IN|22.85|0.836|
> ||+TCN|23.26|0.841|
>
> |Model| Methods|PSNR|SSIM|
> |----|----|----|----|
> | FFA-Net|Original|36.39|0.989|
> ||+IN|35.84|0.987|
> ||+TCN|36.77|0.990|
> | MAXIM| Original|38.11|0.991|
> ||+IN|37.49|0.987|
> ||+TCN|38.54|0.992|
> |C^2PNet|Original| 42.56|0.995|
> ||+IN |42.02|0.994|
> ||+TCN|42.80|0.996|
>
> **3. About extensive experiments.**
>
> The reasons why we conduct the extensive experiments are: (1) Since TCN is proposed to extract lightness (a kind of style) invariant feature while keeping information transition-constant, the extensive tasks are conducted to verify it can benefit more tasks that also related to style information processing. (2) Specifically, for pan-sharpening, it aims to fuse two style images, and we hope TCN can extract their invariant information with information preserving; For medical segmentation, there often exists a domain gap of training and testing sets, and we hope TCN’s robust feature extracting ability can improve the testing accuracy. (3) Besides (2), we have conducted these two tasks before, and other tasks that are related to style information could be benefited from the TCN. We will supplement this discussion in revision.
>
> Although we do not mention testing on downstream applications, which is not our goal and duty. As suggested, we can present related results below, where we perform experiments in DARKFACE on S3FD(following setting in SCI[CVPR 2022])   due to the time limit. We will discuss more in revision.
>
> |Model|Methods|mAP|
> |----|----|----|
> |DRBN| Original|0.559|
> ||+IN|0.574|
> ||+TCN|0.587|
>
> **4. About layout.**
>
> We follow the layout of previous works[2][3], where tables and figures are embedded with words. We will further revise the paper.
>
> [1] Jin et al. Style normalization and restitution for generalizable person re-identification. CVPR 2020.
>
> [2] Antoine et al. Proxy-Normalizing Activations to Match Batch Normalization while Removing Batch Dependence. NeurIPS 2021.
>
> [3] Ekdeep et al. Beyond BatchNorm: Towards a Unified Understanding of Normalization in Deep Learning. NeurIPS 2021.
>
> [4] Zhao et al. Deep symmetric network for underexposed image enhancement with recurrent attentional learning. ICCV 2021.

---

> > ### Comment · Area_Chair_4wye · 2023-08-17
> >
> > Reviewer uwKS, while having a negative rating in the initial review, could you check the Rebuttal and see if the concerns are properly addressed?

---

### Official Review · Reviewer_HV1b · 2023-07-05

**Soundness:** 4 excellent
**Presentation:** 4 excellent
**Contribution:** 4 excellent
**Rating:** 7
**Confidence:** 5

**Summary:**

In this paper, a transition-constant normalization (TCN) is introduced to harness the potential of applying normalization techniques for image enhancement. The TCN is designed with two rules, adhering to the principle of invertible information transmission and dividing the features into two streams with consistent statistical properties. Notably, the TCN is parameter-free and offers multiple usage formats, enabling its integration into existing methods and the creation of new structures. To showcase the benefits of this approach, extensive experiments are conducted across diverse image enhancement tasks.

**Strengths:**

This paper examines the characteristic of normalization and presents its core design approach: "normalizing partial representations to ensure consistent learning while preserving constant information for image reconstruction." Overall, the paper's architecture is clear, and the writing is well-done. The experiments conducted are sufficient. One particularly interesting aspect highlighted in this study is the notion that the introduced TCN serves as a concept rather than a specific operation. The summarized variations of different TCN formats are valuable and open up opportunities for future exploration.

**Weaknesses:**

(a) The characteristic of TCN, which only normalizes partial features, appears to conflict with the motivation. I am concerned that this characteristic might limit its ability. Can you clarify whether this limitation exists?

(b) Layer normalization is widely utilized in transformer structures. Therefore, the paper should include a more comprehensive description and discussion of this aspect.

(c) I am curious why the calculation in Equation 9 is performed in the spatial dimension, similar to IN (Instance Normalization).

(d) It appears that the methods included for TCN are not novel. For instance, why weren't more recent methods like Restormer [1] included in the study?

(e) In the supplementary material provided, it is stated that "the TCN can be constructed in other formats if the invertible rules can be satisfied." It would be beneficial if the paper could provide more detailed information about these alternative formats.

Additional problems:

(a) The introduction and related works sections fail to address invertible techniques, which is a significant oversight.

(b) There are some typos in the paper, such as "other other" in line 155, which should be corrected to "other."

(c) In Figure 2, there is a typo in the calculation of \hat{F_1}. Additionally, the symbol ε is missing from the equation.

(d) The detailed architecture of TCN-Net is not provided in the paper, and it should be included for clarity.

(e) The supplementary material mentions GN, but it does not specify the settings or parameters used for GN. More information about the GN setup should be provided.

[1] SW Zamir. Efficient Transformer for High-Resolution Image Restoration. CVPR 2022.

**Questions:**

If the feature resolution cannot be divided evenly by 2, it may affect the implementation of the TCN deepened on pixel shuffle operation. The pixel shuffle operation typically relies on dividing the feature maps into subgroups and rearranging them to increase the resolution. If the feature resolution is not divisible by 2, it may lead to incomplete subgroups or uneven rearrangement, potentially causing issues in the TCN's implementation. In such a case, the results of the TCN deepened on pixel shuffle operation could be compromised. The network may encounter difficulties in accurately reconstructing images or extracting meaningful features due to the incomplete or uneven reshuffling of feature maps. It is important to consider the compatibility between the feature resolution and the operations used in the TCN architecture to ensure optimal results.

**Limitations:**

Yes. Limitations are adequately addressed.

---

> ### Author Rebuttal · Authors · 2023-08-09
>
> Thanks for your time and efforts and here we present the rebuttal.
>
> **1. Would the TCN’s ability limited?**
>
> The answer is no and here are the reasons: Firstly, the partial normalized features are integrated into the original one evenly in each channel, and thus each channel feature of the TCN’s output is more closer to the normalized feature overall, demonstrating the normalization ability is not limited. Secondly, the toy experiments depict that the TCN can normalize different lightness effectively, meanwhile it can convey information constantly effectively with well self-reconstruct ability. Therefore, the TCN’s ability is not limited both theocratically and experimentally.
>
> **2. More discussion about LayerNorm.**
>
> Thanks for the suggestion. In fact, LayerNorm(LN) plays the role of balance activation function in the Transformer, which is different from what we explore for image enhancement. Besides, LN has a weaker activation ability to the feature than IN, thus improving IN with TCN formats is more meaningful to extend it for normalizing lightness while conveying the information constantly. Finally, the experiments in the supplementary (Table 2) also validates that equipping LN with TCN format can improve performance, and TCN can also improve the performance of NAFNet with LN.
>
> **3. About the calculation in Eq. 9**
>
> This calculation is defined based on IN calculation for the mean and variance, since IN has the best performance for activating features [1] and is often employed for style transfer that is related to image enhancement. Thus, we employ IN for implementing the default TCN format.
>
> [1] Antoine et al. Proxy-Normalizing Activations to Match Batch Normalization while Removing Batch Dependence. NeurIPS 2021.
>
> **4. About performing experiments in recent methods?**
>
> Thanks for your suggestion. We have included a more recent architecture NAFNet in the supplementary (Table 2). Due to the time limit, we also present the results of Restormer as you suggested on the LOL dataset in low-light image enhancement here. Note that the core of our method is to enable existing networks can be improved with the proposed TCN (instead of SOTA), which can also help construct lightweight backbones like the proposed TCN-Net.
> | Model | Methods |PSNR | SSIM |
> |----|----|----|----|
> | Restormer| Original | 22.24 |0.828|
> |        |+IN| 22.06|0.824|
> |        |+TCN | 22.47|0.831|
>
> **5.More information about some technique details in the supplementary.**
>
> Thanks for the suggestion, we describe these details as follows:
>
> (1) TCN can be constructed under other invertible formats. For example, the IN format can be designed following the invertible constraint in [2], which is different from the current format. We will further discuss them in the revision. (2) We have already presented the TCN’s details in supplementary(Sec. 2), we will add a detailed figure to describe it. (3) We set the GN implementation as follows: the group size is set as 4, and each group has the same number of channels. We will add this part in the revision.
>
> [2] Lynton et al. Conditional Invertible Neural Networks for Guided Image Generation. ICLR 2020.
>
> **6. About the discussion of related works and some writing typos.**
>
>   Thanks for your kind reminder. We will add the discussion about invertible techniques, and correct the writing typos correctly in the updated version.
>
> **7.About applying TCN for features with odd number resolution.**
>
>   In experiments, most images’ resolution can be divided by 2, and we usually apply TCN in the large features that their sizes can be divided by 2. When applying TCN on features with odd number resolution, we need to resize the feature to the nearest even resolution preliminary. We will add this technique detail in the updated version.

---

> > ### Comment · Reviewer_HV1b · 2023-08-15
> > **Response to rebuttal**
> >
> > Thank you for your response! After going through the response and the comments of other reviewers, I decided to raise my rating.

---

### Official Review · Reviewer_LSyJ · 2023-07-06

**Soundness:** 3 good
**Presentation:** 3 good
**Contribution:** 4 excellent
**Rating:** 8
**Confidence:** 5

**Summary:**

1. A new normalization technique is introduced that transits information constantly for image enhancement.
2. The introduced TCN covers a range of formats and is compatible with existing method.
3. The introduced technique exhibits extensibility to various scenes and tasks through experiments.

**Strengths:**

It is a novel idea that normalization is improved to meet transition-constant goal with following points:
+ Its design is easy to understand and reasonable
+ Its design demonstrates extensibility in terms of both implementation manners and applicable tasks
+ writing is clear to follow

**Weaknesses:**

However, there remains issues to be dealt with：
- It claims that information transit ability is crucial for image enhancement, why hasn't this ability been incorporated into other operations such as CNN and transformers?
- Its motivation may not be directly correlated with image enhancement, how information transit-inconstant is not supportive for image enhancement? CNN and transformer are not transition-constant and they are commonly used and have proven effective.
- Its built manner is limited in few cases, when for implementation, small resolution feature may not be operated with resolution need to be further reduced.
- It lacks of visual comparison and deep analysis about ablation studies, merely providing quantitative results could not be very supportive.
- Its toy experiments may not be supportive. When for self-reconstruction, TCN still has obvious lower performance than with “None”.
- Its design is similar with Half-instance normalization, yet the performance is not significantly higher than Half-instance normalization in ablation studies.
- It contains writing errors, inaccurate equation denotations and figure denotations. Please check and rectify them carefully.

**Questions:**

Why the extensive experiments are performed in guided image super-resolution task and medical segmentation task? What about other tasks? Besides, what makes this technique is able to be extended to other tasks? What makes authors consider to do this thing? Furthermore, Why not incorporate more kinds of models (CNN or transformer) as baselines?

**Limitations:**

The limitations and potential negative societal impact of their work have been addressed in this paper.

---

> ### Author Rebuttal · Authors · 2023-08-09
>
> Thanks for your time and efforts and here we present the rebuttal.
>
> **1. About the relationship of transition-constant ability and image enhancement.**
>
> First, for architectures such as CNN and transformer, they convey information with the driven of reconstruction losses, thus the whole part of them is nearly to transition-constant. However, for Instance normalization (IN), it discards information [1] instead of learning to filter the information like CNN or transformer, making the sub-sequential parts cannot restore the information. This is how IN’s weakness is different from other architectures. Second, due to the IN’s weakness, and its advantage is normalizing different lightness and thus is useful for image enhancement tasks, we improve the IN with the transition-constant ability that enables IN can be practically used for image enhancement. Note that this ability is equipped for IN instead of other parts. Finally, DSN [2] shows the transition-constant ability of networks can be developed for image enhancement, while this work doesn’t explore IN’s property. We will add the above discussion in the revision.
>
> [1] Jin et al. Style normalization and restitution for generalizable person re-identification. CVPR 2020.
>
> [2] Zhao et al. Deep symmetric network for underexposed image enhancement with recurrent attentional learning. ICCV 2021.
>
> **2. About applying TCN in small features.**
>
>    We implement TCN on the feature with large size features, while small size features contain much less information than the large size, therefore applying the TCN in such information-reduced feature is not necessary. We will further clarify this point in the revision.
>
> **3. About the demonstration of ablation studies.**
>
> Firstly, we need to emphasize that the comparison results of baseline and with IN itself can be the ablation, therefore, the number of results is enough. Besides the ablation studies in Table 8 of the supplementary, other parts can also depict this role. The analysis of toy experiments in the supplementary verifies the reasonableness of our TCN’s design, which corresponds to the results of Table 8. Due to limited space, we will supplement more analysis and visual results in the revision.
>
> **4. About the toy experiment.**
>
> We conduct the toy experiment to show that the TCN has better performance than IN for conveying information. In fact, as depicted in Fig.4, inserting TCN for self-reconstruction has much higher performance than inserting IN, and the result of “None” is a reference to measure how the TCN is effective for conveying information (TCN is only 4.08 dB lower than “None”, while 21.95 dB higher than IN, note that 4dB is not a large number in 50+dB level). Therefore, the results of toy experiments can demonstrate better transition-constant properties of TCN than IN. Besides, the toy experiment also depicts the normalization ability of TCN.
>
> **5. About the half-instance normalization.**
>
> Firstly, our TCN outperforms it by about 0.2dB, which is considerable.  Then, half-instance normalization is an ablation module without theocratic design, while the proposed TCN is designed with strong motivation based on the transition-constant constraint, which is a supplement of half-instance normalization. Notably, our method would inspire more work toward designing normalization formats that is suitable for image enhancement tasks and other image processing tasks, which have not been mentioned or explored before.
>
> **6. Some suggestions about writing.**
>
> Thanks for your kind reminder, we will revise our paper carefully in the revision.
>
>
> **7. About extensive experiments.**
>
> **(1)** The reasons why we add the extensive tasks: Since TCN is proposed to extract lightness (a kind of style) invariant feature while keeping information transition-constant, we introduce the extensive tasks to verify this property can benefit more tasks that are similar to image enhancement, where these tasks are related to style information processing, thus proving the TCN’s extensive ability.
> **(2)** The reasons why we choose these two tasks: These two tasks are relevant to the style of information processing and thus can evaluate the TCN’s extensibility. For pan-sharpening, it aims to fuse two modality information of different style, and we hope TCN can extract their invariant information with information preserving; For medical segmentation, there often exist a domain gap of training and testing, and we hope TCN’s robust feature extracting ability can improve the testing accuracy that could be influenced by the domain gap. Moreover, we have conducted these two tasks before, and some other tasks can also be explored with TCN.
> **(3)** Discussion about more tasks: Yes. We believe that the TCN can also be extended to other tasks about style processing, such as domain adaptation and multi-sensor image fusion. The TCN could extract the invariant component of different style information effectively that elevate the performance.
> **(4)** Lastly, we have presented CNN- and transformer-based backbone with TCN in the supplementary materials, proving the effectiveness of the TCN. We will add the above information in the revision properly.

---

> ### Comment · Area_Chair_4wye · 2023-08-18
> **Please respond to the rebuttal**
>
> Please check if your concerns have been addressed, and justify your final decision.
>
> Best,
>
> AC

---

> ### Comment · Reviewer_LSyJ · 2023-08-19
>
> I am satisfied with the explanation in the rebuttal, which resolves my concerns. Considering the technique contribution of this paper, I recommend accepting this paper.

---

### Decision · Program_Chairs · 2023-09-21

**Decision:**

Accept (spotlight)

**Comment:**

This paper introduce transition-constant normalization (TCN) to harness the potential of applying normalization techniques for image enhancement. The scores from all reviewers are positive and most reviewers find the approach interesting and the results are convincing. The AC agrees with the reviewers on accepting the paper.